# An Empirical Investigation of Domain Generalization with Empirical Risk Minimizers

**Ramakrishna Vedantam**
FAIR, New York
ramav@fb.com

**David Lopez-Paz**[*]
FAIR, Paris
dlp@fb.com

**David J. Schwab**[*]
ITS, CUNY Grad Center
FAIR, New York
davidjschwab@gmail.com

## Abstract

Recent work demonstrates that deep neural networks trained using Empirical Risk Minimization (ERM) can generalize under distribution shift, outperforming specialized training algorithms for domain generalization. The goal of this paper is to further understand this phenomenon. In particular, we study the extent to which the seminal domain adaptation theory of Ben-David et al. (2007) explains the performance of ERMs. Perhaps surprisingly, we find that this theory does not provide a tight explanation of the out-of-domain generalization observed across a large number of ERM models trained on three popular domain generalization datasets. This motivates us to investigate other possible measures—that, however, lack theory—which could explain generalization in this setting. Our investigation reveals that measures relating to the Fisher information, predictive entropy, and maximum mean discrepancy are good predictors of the out-of-distribution generalization of ERM models. We hope that our work helps galvanize the community towards building a better understanding of when deep networks trained with ERM generalize out-of-distribution.

## 1 Introduction

Conventional wisdom in domain generalization was recently upturned by the work of Gulrajani & Lopez-Paz (2020) on the DomainBed benchmark. In their work, the authors use careful model selection and evaluation to show that models trained using empirical risk minimization (Vapnik, 1999) are able to achieve near state-of-the-art performance on a variety of popular benchmarks, surpassing the performance of most algorithms specialized for domain generalization (Arjovsky et al., 2019; Yan et al., 2020; Li et al., 2017a).

In a parallel research effort, there has been an increased interest in understanding when deep neural networks trained using empirical risk minimization generalize well in-domain (Jiang et al., 2019; Neyshabur et al., 2017; Dziugaite & Roy, 2017; Bartlett et al., 2017; Arora et al., 2018; Daniely & Granot, 2019). However, understanding when empirical risk minimizers are able to generalize out-of-distribution remains poorly understood. Answering this question would help us characterize the failure modes of domain generalization algorithms, and to develop specialized baselines that would allow us to break through the dominance of empirical risk minimization.

We argue that the out-of-distribution generalization performance of a classifier depends mainly on two factors. On the one hand, we expect good out-of-distribution generalization only for classifiers able to achieve good in-distribution generalization (also called source-domain validation accuracy). On the other hand, we expect out-of-distribution generalization to degrade as the discrepancy between the distributions of training and testing data increases, relative to the function family of interest. This reveals an interplay between the properties of the neural network, the used training data, and the

---

[*] Both D.L.P and D.J.S. contributed to the work equally
35th Conference on Neural Information Processing Systems (NeurIPS 2021).

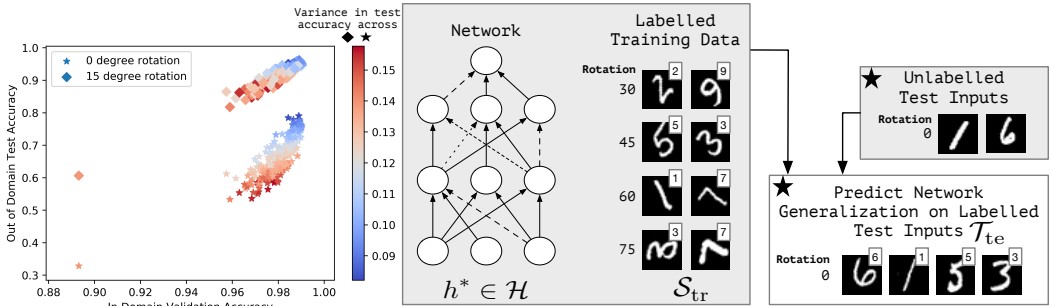

Figure 1: **Left:** Source-domain validation accuracy (x-axis) *v.s.* Novel, Target-domain test accuracy (y-axis) for a set of 100 models trained on RotatedMNIST environments with rotations between 30-75 degrees. Notice that the ability to generalize to novel domains can vary substantially (0 degrees, ★ vs 15 degrees, ♦). **Right:** We seek to characterize the conditions under which a network generalizes to unseen domains, given unlabelled examples from the novel, test domain, the weights of the network trained via empirical risk minimization on the source (training) data and the training data.

differences between training and testing regimes (Figure 1). Given all of these distinct ingredients, how does one build a coherent theory of domain generalization with strong predictive power for modern, over-parametrized neural networks?

The seminal work of Ben-David et al. (2007, 2010) lays down a theoretical foundation for the related setup of unsupervised domain adaptation. In particular, the theory in Ben-David et al. (2007, 2010) upper-bounds the out-of-distribution performance of an ERM based on a fixed feature space as the sum of two terms: (i) the source-domain validation accuracy, and (ii) the distance between the input distribution of source-domain and target-domain examples under the fixed feature space. This theory has found success in practice, improving both linear models (Ben-David et al., 2007) and deep neural networks (Ganin et al., 2015) in the setup of unsupervised domain adaptation. Algorithmically speaking, one can implement this theory by finding a feature representation under which (i) one can perform well on source-domain inputs, and (ii) it is not possible to distinguish source-domain inputs from target-domain inputs.

While the theory above deals with the setup of unsupervised domain adaptation (where algorithms have access to unlabeled target-domain data during training), here we are interested in the related but distinct domain generalization scenario where algorithms only have access to source-domain data during training. Therefore, to remain in the domain generalization setup, we will put the theory of Ben-David et al. (2007) to use from a different angle. Instead of training representations in such a way that encourages source-domain and target-domain inputs to be indistinguishable, our methodology consists in training a very large number of ERM models only using source-domain data, to later investigate if the models exhibiting good out-of-distribution generalization indeed learned a feature space under which it is difficult to tell apart source-domain and target-domain inputs.

Training over 12,000 deep neural network image classifiers using empirical risk minimization, using the standard DomainBed pipeline, we reveal that the theory of Ben-David et al. (2007) has limited power to predict when a classifier will generalize out-of-domain. We investigate this further by unpacking whether assumptions in the theory are violated or if the bounds are slack. Our findings suggest that in practice, while the assumptions from the theory are largely not violated in the learned feature spaces of ERMs, one can obtain good generalization despite being able to tell apart the source and target data, limiting the practical applicability of the theory to explain domain generalization. We investigate and provide more intuitions for this phenomenon in section 4.

In search of other correlates of domain generalization that might point the way to new theoretical approaches, we set out to build a catalog of measures that could predict the out-of-distribution generalization performance of our suite of ERM trained neural networks. In building such a catalog of generalization measures, we take inspiration from recent literature undertaking a similar effort to explain variance of *in-domain* generalization of deep neural networks (Jiang et al., 2019; Neyshabur et al., 2017; Dziugaite et al., 2020). After an exhaustive empirical evaluation, we find that several of the proposed measures are relatively strong predictors of out-of-distribution performance. These include measures related to predictive entropy, the Mixup criterion (Zhang et al., 2017), the norm of the Jacobian (Novak et al., 2018), and the Fisher information matrix (Amari, 1998).

To summarize, our work makes the following contributions:

- We perform a large scale empirical study testing the theory from Ben-David et al. (2007, 2010) on deep neural networks trained on the DomainBed (Gulrajani & Lopez-Paz, 2020) domain generalization benchmark.

- We find that the theory has limited ability to predict which ERMs will generalize out-of-distribution.

- We perform an empirical evaluation of several other candidate measures for explaining domain generalization of neural networks, finding measures that outperform theoretically motivated quantities.

## 2 Background on Relevant Theory

There is a rich line of work on the theory of domain adaptation, which focuses on learning and adapting when presented with test data drawn from a different input distribution from that seen during training (Ben-David et al., 2007; Mohri & Medina, 2012; Cortes et al., 2019). There are also a number of algorithms for domain adaptation which make use of unlabeled data to learn more invariant predictors, both for classical kernel machines (Pan et al., 2011; Gretton et al., 2009) as well as deep neural networks (Ganin et al., 2015). The results typically provide bounds on the test error in a target domain in terms of the test error of the model on a training domain, and a measure of the difference between the train and target input distributions in a feature space. Here we are interested in whether the theory can accurately predict out-of-distribution performance of an ERM.

In particular, we first focus on a classic result from Ben-David et al. (2007), which, given a fixed feature representation ($\mathcal{R}$), derives generalization bounds for the test performance in terms of how well the model can distinguish source from target samples. This work focuses on the single-source (SS) setting, where a single source-domain is used for training. A central result from this work is a bound on the test error on the target domain ($\epsilon_T(h)$), consisting of three distinct terms: the source-domain validation error ($\epsilon_S(h)$), the so-called $\mathcal{H}$-divergence between the source and target domains, and the joint source-target validation error when optimizing a single model for both environments, i.e. $\lambda = \min_{h \in \mathcal{H}} \{\epsilon_T(h) + \epsilon_S(h)\}$. In fact, in the theory, $\lambda$-closeness is assumed, and we will test this assumption later. We consider the predictive value of each of these three terms separately as well as their sum in our analysis. The $\mathcal{H}$-divergence can be understood as related to and approximated by the best performance of a classifier in the function family $\mathcal{H}$ trained to distinguish samples from the source and target domains.

Subsequent work generalized this theory to multi-source (MS) training in Ben-David et al. (2010). Assuming an equal weighting of all training domains, the gap between the test errors in the target and source domains can be bounded above by the sum of two terms: the average $\lambda_j$, where $\lambda_j = \min_{h \in \mathcal{H}} \{\epsilon_T(h) + \epsilon_j(h)\}$, and the $\mathcal{H}\Delta\mathcal{H}$-divergence, similar to the $\mathcal{H}$-divergence but for the symmetric difference hypothesis class. A full description of these terms can be found in the supplementary material.

## 3 Experiments

In this section we describe our methodology and experimental setup for addressing the problem of implicit domain generalization of ERMs. We first start with a brief overview of the DomainBed benchmark and then describe the collection of models we train in an effort to measure generalization.

### 3.1 DomainBed Benchmark and Training Setup

DomainBed (Gulrajani & Lopez-Paz, 2020)[1] is a recently released toolbox for domain generalization which provides implementations of various domain generalization algorithms such as Invariant Risk Minimization (Arjovsky et al., 2019), Variance Risk Extrapolation (Krueger et al., 2020), and meta-learning based domain generalization (Li et al., 2017a) in addition to unsupervised domain adaptation approaches (Sun & Saenko, 2016; Ganin et al., 2015; Yan et al., 2020). It also provides various datasets such as RotatedMNIST (Ghifary et al., 2015), VLCS (Fang et al., 2013), and PACS (Li et al., 2017b). These datasets are provided in the form of multiple environments $e = \{(X_e^i, y_e^i)\}_{i=1}^N$, each

---

[1] https://github.com/facebookresearch/DomainBed (MIT License)

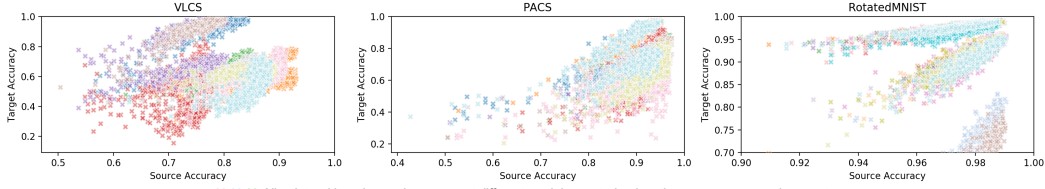

Figure 2: **Target OOD accuracy** *v.s.* **Source Validation Accuracy** for VLCS, PACS, and RotatedMNIST (left to right). Each datapoint corresponds to a trained model (out of 12 thousand) using ERM and a test environment, and each set of training environments has a different color (*e.g.*, in fig. 1 all models trained with rotations 30-75 on RotatedMNIST are assigned a given color).

with a different training distribution $p_e(X)$ and (generally) the same conditional label distribution $p(y|X)$. Domain generalization algorithms are then trained on a subset of source environments $E_{\text{source}} = \{e_{\text{source}}^j\}_{j=1}^M$ and tested on target environments $E_{\text{target}} = \{e_{\text{target}}^j\}_{j=1}^K$.

An intriguing result from the DomainBed benchmark is that standard ERM training can match or outperform more specialized domain generalization algorithms. We would like to understand this phenomenon better by characterizing properties that distinguish ERMs (Empirical Risk Minimizers) which perform well from those which do not. We follow Gulrajani & Lopez-Paz (2020) in combining all the training environments into one source dataset $S = \bigcup_{e \in E_{\text{train}}} e$ used to train our ERMs. Testing is performed independently on each target environment.

We use the same network architectures as those used in the DomainBed paper (see supplemental information for more details). Overall our networks span shallow convolutional neural networks with 386K parameters (on the MNIST datasets) to residual networks with 25 million parameters (on VLCS and PACS). We prioritize a careful, exhaustive hyperparameter sweep with the above architectures, to obtain models with different levels of performance (Gulrajani & Lopez-Paz, 2020), over studying different architectures which is arguably a second order effect on this problem. Following Gulrajani & Lopez-Paz (2020), for each combination of dataset and training environments, we pick 100 random hyperparameter settings of batch size, learning rate, weight decay, and dropout (for resnet models). For both source $S$ and target $T$, we hold out 50% of the data for validation. Since no training happens on $T$ this means that we can use 50% of the data in $T$ for computing generalization measures and the other 50% for assessing the out-of-distribution error of an an empirical risk minimizer $\hat{h}$, which we term as $\epsilon_T(\hat{h})$. Analogously, we denote the validation error of the model on the source domain as $\epsilon_S(\hat{h})$.

### 3.2 Trained Empirical Risk Minimizers (ERMs) and Implicit Domain Generalization

Using the above procedure, we train approximately 12,000 models on a compute cluster with Volta GPUs using PyTorch (Paszke et al., 2019). All models are trained for 5000 training steps and the model saved at the last step is used for analysis (Gulrajani & Lopez-Paz, 2020). fig. 2 shows the results of the sweep. We note that the source-domain validation accuracy or error $\epsilon_S(\hat{c})$ is a good candidate for explaining performance on domain generalization (see fig. 2). This is a common practice in the literature (Hendrycks et al., 2020), where model selection is often performed based on this criteria. However, we note that for a given source-domain validation accuracy, there is significant variation in the corresponding out-of-distribution accuracy, suggesting need for measures to characterize generalization more tightly.

## 4  Testing the Predictive Power of Theory-based Measures

In this section, we empirically evaluate various elements of the theory discussed in section 2 on 12,000 neural networks trained via ERM on the DomainBed benchmark.

Our general methodology for testing the theory involves two major components. Firstly, we estimate the tightness in the bounds from Ben-David et al. (2007) empirically, and secondly, we study if the terms from the theory are predictive of performance. As explained in section 2, given $\tilde{D}_S$, the set of features extracted on a source domain, and $\tilde{D}_T$, the set of features extracted on the target domain, the

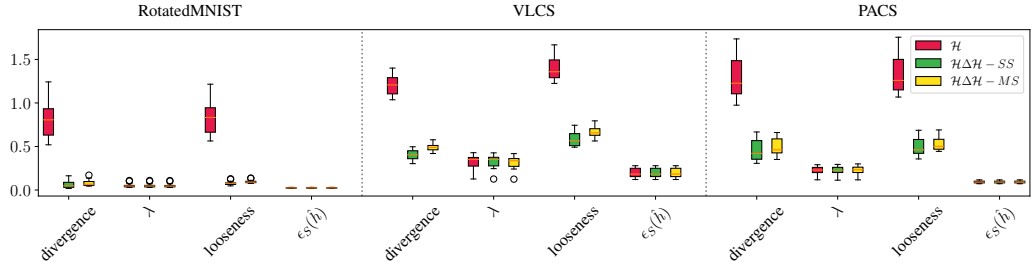

Figure 3: Different DomainBed datasets (columns) showing the divergence measure, $\lambda$ – indicating how close source and target domains are, $\epsilon_S(\hat{c})$– the source domain validation error and looseness of the bounds (Ben-David et al., 2007, 2010) computed empirically for bounds relating to ($\mathcal{H}$-divergence, $\mathcal{H}\Delta\mathcal{H}$-Single Source (SS), $\mathcal{H}\Delta\mathcal{H}$-Multi Source (MS), in legend). Each figure shows box-plots of the distributions of the quantities (across 3K trained models for PACS and VLCS, and 6K for RotatedMNIST). **y-axis** for each plot is in units of error.

theory provides bounds on the target error $\epsilon_T(c)$ in terms of the source error $\epsilon_S(c)$, $\mathcal{H}$-divergence $(d_{\mathcal{H}}(\tilde{D}_S, \tilde{D}_T))$ and closeness of the domains ($\lambda$) (both computed in the feature space of a model):

$$\epsilon_T(c) \le \epsilon_S(c) + d_{\mathcal{H}}(\tilde{D}_S, \tilde{D}_T) + \lambda, \tag{1}$$

**Estimation Considerations:** In order to test how well the theory works in practice, we need to estimate the $\mathcal{H}$ and the $\mathcal{H}\Delta\mathcal{H}$-divergences from finite samples. We construct and empirically study the variance of estimators for both in the supplementary material, finding that estimates of $\mathcal{H}\Delta\mathcal{H}$-divergence have a lower SNR than $\mathcal{H}$-divergence (for which there are known convergence guarantees when $\mathcal{H}$ has a finite VC-dimension). However, for completeness we report results with both.

**How loose are the bounds?** In fig. 3 we plot for each divergence measure and dataset, the right hand side of the bound in eq. (1). This reveals several interesting observations. Firstly, when the bounds, which are of the form "divergence measure $+ \lambda + \epsilon_S(\hat{c})$" (see eq. (1) for an example) are loose (first box-plot in each figure) (where looseness is defined as the difference between right hand side and left hand side of eq. (1)) it is not because the representations $\mathcal{R}$ entailed by the networks do not yield domains which are $\lambda$ close but because the divergence between the source and target domains is large. Thus, in principle ERMs seem to yield feature spaces from which one can support the learning of both target and source domain labels. However, ERMs do not appear to learn feature spaces which implicitly align the source domain to the (unseen) target domain in the domain generalization setting. In unsupervised domain adaptation (uDA) one could enforce the divergence to be small (Ganin et al., 2015) (since uDA assumes access to the target domain inputs $\tilde{D}_T$), which could result in tighter bounds on test error, as long as the domains remain lambda-close. However, recent work (Zhao et al., 2019; Wu et al., 2019) shows that enforcing a low divergence can yield representations which are no longer lambda close, rendering the theory less useful in practice from an uDA perspective. Thus, having a low divergence seems neither sufficient nor necessary for generalization.

Our goal is not just to understand how tight or loose the bounds are, but to check how well the measures computed based on them correlate with target-domain test error. What governs the regimes in which the bounds correlate well with test error $\epsilon_T(\hat{h})$? Do tighter bounds lead to stronger correlation with test error? We find that this is not necessarily the case, for example, on RotatedMNIST with $\mathcal{H}\Delta\mathcal{H}$-divergence-MS based bound ($\mathcal{H}\Delta\mathcal{H}$-divergence-MS $+ \lambda + \epsilon_S(\hat{c})$) (where the gap is very low (fig. 3, right, yellow-boxes) the Spearman's $\rho$ with observed generalization $\epsilon_T(\hat{h})$ is 0.422 while on VLCS it is 0.567. Thus, it is not just how biased upper bound is, but also the variance which can play a major role in predicting generalization.

**Why do variations in $\mathcal{H}$-divergence not correlate well with observed generalization despite $\lambda$-closeness?** We illustrate with a simple example (fig. 4) how one can obtain good generalization despite being very far from the source domain in terms of $\mathcal{H}$-divergence. In such a regime, one can obtain a high $\mathcal{H}$-divergence even when the classifier learnt on the source domain generalizes well to the target domain. Of course, the theory accounts for this in principle, since the bound on the target error is not violated. However, it is not useful in practice for predicting generalization despite the $\lambda$-closeness assumption being satisfied.

**Is the theory useful for predicting generalization?** In order to use the bounds in a more practical setting, we must recall that we do not have access to the test labels from the target domain, and thus cannot compute $\lambda$-closeness. Instead, we consider a biased estimate of the bound, namely $\epsilon_S(\hat{h}) +$

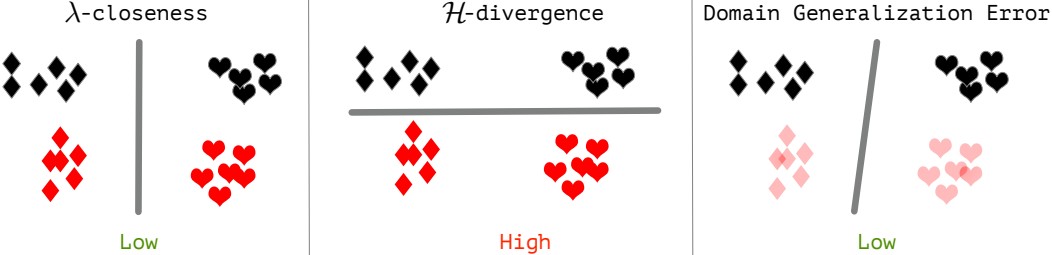

Figure 4: Consider a binary classification problem between ♦ and ♥ in a 2D space and a family of linear classifiers $\mathcal{H}$. **Black** points correspond to the source domain and **red** points correspond to the target domain. Our goal in domain generalization (right) is to train a classifier on the source domain black points and generalize well to the test domain (shaded red points). The datasets are $\lambda$-close (left) since there exists a classifier in $\mathcal{H}$ which does well on the union of the source and target domains. Further, the distributions are far in terms of $\mathcal{H}$-divergence since a linear classifier can easily separate the source data points from the target datapoints (middle). However, the generalization gap (right) is low, suggesting how one might find poor correlations between $\mathcal{H}$-divergence and target error $\epsilon_{T(c)}$, despite $\lambda$-closeness from Ben-David et al. (2007) being satisfied.

$\mathcal{H}$-divergence, for the $\mathcal{H}$-divergence based measures as an illustration. Note that one can understand $\lambda$-closeness as an assumption in the original theory, as opposed to something that is intended to be empirically computed. Thus, in this section we compute practically accessible components of the theory and measure correlation with actual error. Overall, we find that the Spearman's $\rho$ across conditions drops substantially without accounting for $\lambda$-closeness. For example, for $\mathcal{H}\Delta\mathcal{H}$-divergence-MS based bound the Spearman's $\rho$ with generalization drops from 0.496 to 0.136 on removing $\lambda$-closeness. Similar trends can be seen for $\mathcal{H}\Delta\mathcal{H}$-divergence-SS on removing $\lambda$ from the bound, which drops from 0.509 to 0.152 and $\mathcal{H}$-divergence which goes from 0.215 to 0.027. In a sense, this is not very surprising, since $\lambda$-closeness uses the target-domain labels which provides significant signal for predicting the target domain error. However, this indicates that the theory cannot be used to great effect for predicting generalization in practice. Finally notice that among these measures, one performs much better generally when using $\mathcal{H}\Delta\mathcal{H}$-divergence-MS or $\mathcal{H}\Delta\mathcal{H}$-divergence-SS than when using $\mathcal{H}$-divergence.

## 5 Exploring Empirical Measures

We have seen that the classic theory of domain generalization does not succeed at providing measures that are strongly predictive of out-of-distribution generalization ($\epsilon_T(\hat{c})$) in practice. Thus we next turn to exploring new measures that may provide better predictive power. We emphasize that our goal here is not to formulate highly predictive measure for use in practice, but rather to point towards promising directions for future theoretical investigations.

In pursuit of these goals, given the heterogeneity of the environments (see fig. 2), we make the following methodological choices: (1) we seek good predictors of generalization for each combination of dataset and environment (which we term a *condition*) and aggregate across conditions to find measures which are reliable across different conditions, (2) we learn both separate regressors for each condition and a single regressor across all conditions. In all cases, we perform Ridge regression with leave-one-out cross validation for all the analysis. Across all regressions, we use a train, test split of $(0.7, 0.3)$. We use the coefficient of determination $R^2$ as a metric which indicates the fraction of the variance of the target variable explained by the regression, and report the average $R^2$ of a predictor across all conditions of dataset/environment pairs. Following the methodology of Jiang et al. (2019), we also compute and emphasize Spearman's rank correlation coefficient (Spearman's $\rho$) to quantify how well each measure predicts which ERMs will generalize out-of-distribution.

### 5.1 Description of selected measures

Here we describe select measures that quantify the degree to which the model is sensitive to changes in the distribution of inputs. Here we briefly summarize a few of the most performant measures we find. A full list can be found in the supplemental information, including variants of many measures. First, we briefly introduce measures based on the Fisher information of the model and how it changes

when computed on samples from the source or target environment. Then we study the norm of the Jacobian of the model computed on source or target environment samples. Finally, we introduce a measure based on the entropy of the model's predictions on source and target samples.

**Fisher Based Measures.** The Fisher information (Amari, 1998) measures the sensitivity of model outputs to changes in the parameters. We expect that models will generalize well out-of-distribution when the parameter directions that affect model outputs are similar when computed on train and test input samples. Let the model output for class $i$ on an input sample $x_n$ be $h_{\theta,i}(x_n)$, where $\theta$ denotes the model parameters. The Fisher information is then defined as $I = \frac{1}{N} A^T A$, where $A_{\mu,(in)} = \frac{1}{\sqrt{h_{\theta,i}(x_n)}} \frac{\partial h_{\theta,i}(x_n)}{d\theta_\mu}$. Our generalization measures based on the Fisher information compare the Fisher obtained on the source training dataset $I_{\mathcal{S}_{\text{tr}}}$ to the Fisher obtained on the target data $I_{\mathcal{T}_{\text{tr}}}$. Note that the Fisher, unlike the Hessian does not require labeled data and thus can be computed on the unlabeled images. For details, see the supplementary material.

**Jacobian Norm Based Measures.** The input-output Jacobian, a measure of sensitivity of the model output to changes in the input, is defined as $J_{i,\alpha} = \frac{\partial h_i(x)}{\partial x_\alpha}$. Intuitively, a model that is less sensitive to changes in the input will generalize better. Thus, it has been studied in the context of robust learning (Hoffman et al., 2019) and in-distribution generalization (Novak et al., 2018), where it was found to be predictive of generalization at the level of individual test points. Similar to Novak et al. (2018), we compute the Frobenius norm of the Jacobian. See supplementary material for full details.

**Mixup Based Measures.** Mixup (Zhang et al., 2018) was proposed as a more robust alternative to ERM that encourages the learned function to be smooth, and has been shown to smooth the network's input-output Jacobian (Carratino et al., 2020). Here we use mixup as a generalization measure, and use the model's score function $h$ instead of the labels $y$ for interpolation. Our motivation is the same as that for the Jacobian: if the function is less smooth around target examples, the network should not generalize as well. Given a neural network $h(X)$, a dataset $\mathcal{T}$, and $\lambda \sim \text{Beta}(\alpha, \alpha)$, we compute the Mixup measure as: $\frac{1}{|\mathcal{T}|} \sum_{X_i \in \mathcal{T}, X_j \in \mathcal{T}; i \neq j} \left( \lambda h(X_i) + (1-\lambda)h(X_j) - h(\lambda X_i + (1-\lambda)X_j) \right)^2$. See supplemental material for a detailed algorithmic description for computing this measure.

**Entropy on Target Data.** We also compute the output-entropy of the neural network $h$ on the target domain data $\mathcal{T}_{tr}$ (Entropy-Target): $\frac{1}{|\mathcal{T}_{\text{tr}}|} \sum_{X_i \in \mathcal{T}_{\text{tr}}} \sum_{j=1}^{\mathcal{Y}} -\log\left(h(X_i)[j]\right) \cdot h(X_i)[j]$. Entropy-Source is defined analogously but on source domain samples.

## 5.2 Results with Single Measures

We provide a representative sampling of the results for the measures we study in this section, while deferring to the appendix for more detailed results spanning all the measures that we study. For this section, we first computed the Spearman's $\rho$ for each of the measures with target-domain error, $\epsilon_T(\hat{c})$, and selected the measure from each family of related measures (say, those related to the Jacobian for example) which has the highest correlation. As a pre-processing step, we canonicalize the measures to always have a positive correlation with $\epsilon_T(\hat{c})$ (see appendix for more details).

**Which measures outperform the theory based measures $\mathcal{H}\Delta\mathcal{H}$-divergence-SS and $\mathcal{H}$-divergence?** In terms of Spearman's $\rho$, we find that Entropy-Source (0.598), Entropy-Target (0.734), Jacobian (0.420), MMD-Gaussian (0.283), and Fisher Align (0.441) all outperform the theory inspired measures from section 4, where the best performing measure was $\mathcal{H}\Delta\mathcal{H}$-divergence-SS at Spearman's $\rho$ of 0.152 table 1. Intriguingly, Entropy-Target (0.734) outperforms in-domain test error, $\epsilon_S(\hat{c})$, (0.712) in terms of Spearman's $\rho$, while using $R^2$ it is slightly worse (0.608 *v.s.* 0.582).

**Practically analyzing the performance of Entropy-Target.** We next analyze the performance of Entropy-Target across the different datasets. We find that Entropy-Target outperforms $\epsilon_S(\hat{c})$ on the RotatedMNIST dataset (0.853 *v.s.* 0.744 Spearman's $\rho$), while on the VLCS and PACS datasets $\epsilon_S(\hat{c})$ is better, with 0.625 *v.s.* 0.557 Spearman's $\rho$ on VLCS and 0.732 *v.s.* 0.662 Spearman's $\rho$ on PACS. One plausible explanation for the strong performance in this regime is shown in fig. 5. Consider the case where we train with environments with rotation 30-75 degrees in RotatedMNIST (as in fig. 1). In such a situation $\epsilon_S(\hat{c})$ cannot distinguish between the expected performance on the 0 degree and 15 degree test environments.

Table 1: Different measures (rows) evaluated on either Spearman's $\rho$ or $R^2$ metric between the measure and $\epsilon_T(\hat{c})$ (columns) where each measure is either used **Alone** as a predictor or in conjunction with in-domain test error, $\epsilon_S(\hat{c})$, and is allowed to use only one set of parameters across all datasets and source environments (**Joint**) or allowed to set parameters differently for different environments (**Stratified**). We report per-environment correlation averaged across environments. Approaches which perform better than source-domain validation accuracy are shown in **bold**, while approaches which perform better than the theory based measures are highlighted (p-values for all the results below are orders of magnitude lower than performance numbers, and are thus dropped).

| | Spearman's $\rho$ | | | $R^2$ | | |
|---|---|---|---|---|---|---|
| | | With $\epsilon_S(\hat{c})$ | | | With $\epsilon_S(\hat{c})$ | |
| | Alone | Joint | Stratified | Alone | Joint | Stratified |
| $\epsilon_S(\hat{c})$ | 0.712 | - | - | 0.608 | - | - |
| $\mathcal{H}$-divergence (train)-MS | 0.151 | 0.36 | 0.753 | 0.115 | 0.277 | 0.656 |
| Entropy-Source | 0.598 | 0.712 | 0.705 | 0.469 | 0.555 | 0.62 |
| Entropy-Target | **0.734** | **0.777** | 0.812 | 0.582 | 0.623 | 0.710 |
| Fisher Align | 0.441 | 0.092 | 0.727 | 0.284 | 0.098 | 0.626 |
| $\mathcal{H}\Delta\mathcal{H}$-divergence (train)-SS | 0.154 | 0.336 | 0.772 | 0.187 | 0.214 | 0.662 |
| Jacobian | 0.420 | **0.762** | 0.834 | 0.293 | 0.610 | 0.741 |
| Mixup | 0.279 | 0.515 | 0.749 | 0.172 | 0.418 | 0.651 |
| MMD-Gaussian | 0.283 | 0.560 | 0.829 | 0.260 | 0.444 | 0.766 |
| MMD-Mean-Cov | 0.0998 | 0.733 | 0.808 | 0.141 | 0.565 | 0.710 |
| $L_2$-Path Norm. | 0.39 | 0.699 | 0.712 | 0.090 | 0.553 | 0.615 |
| Sharpness | 0.117 | 0.710 | 0.713 | 0.206 | 0.552 | 0.624 |

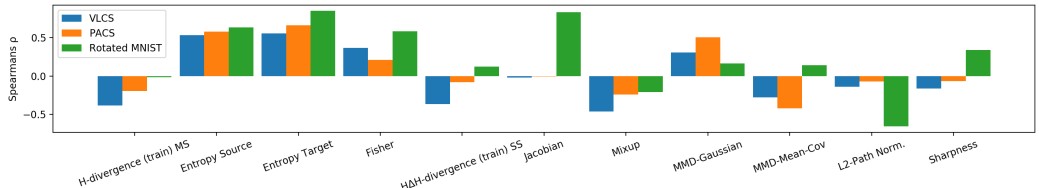

Figure 6: Spearman's $\rho$ (y-axis) against different measures (x-axis) for the different datasets.

However we see in fig. 5 that the entropy distributions across different models computed on the 0 and 15 degree target environments are quite different, with higher confidence on the 15 degree test environment where better generalization occurs.

**What contributes to the performance of Entropy-Source?** We find that across all models, the Entropy-Source (computed on the source training set) is highly correlated with $\epsilon_S(\hat{c})$ (e.g. Spearman's $\rho$ of 0.857 on RotatedMNIST). This suggests that the performance of Entropy-Source is likely because of its relationship to $\epsilon_S(\hat{c})$. This also potentially explains why using both $\epsilon_S(\hat{c})$ and Entropy-Source jointly table 1 has the same performance as using $\epsilon_S(\hat{c})$ alone (0.712 Spearman's $\rho$).

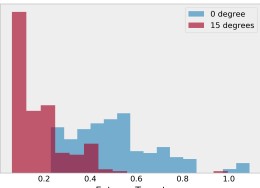

Figure 5: Entropy distributions for targets with 0 and 15 degree rotation

**Performance of measures split across datasets.** In fig. 6, we find that for many measures the correlation has the same sign across the different datasets, which is encouraging and indicates that the proposed measures might be useful to understand generalization. Overall we find that Entropy-Source, Entropy-Target, $\mathcal{H}$-divergence (train), Fisher Align, Mixup, $L_2$-Path Norm, MMD-Gaussian are reliable in terms of the sign of the correlation. However, $\mathcal{H}$-divergence (train) and $L_2$-Path Norm have a high variance across datasets, which is not ideal.

**Do Jacobian or Fisher measures perform better?** While both Jacobian and Fisher Align have similar Spearman's $\rho$ values overall (around ∼ 0.4) we find that the Jacobian, while highly performant

on RotatedMNIST with Spearman's $\rho$ of 0.83 across environments, performs quite poorly on PACS and VLCS (Spearman's $\rho \sim 0$), and thus may not be a reliable measure. Previous work has utilized the Jacobian (Novak et al., 2018) on four different image datasets (CIFAR10, FashionMNIST, MNIST, CIFAR100) and found it to be predictive of generalization at the level of individual test points coming from the same distribution. However, our results illustrate that in the out-of-distribution setting, the Jacobian is less reliable. In contrast, the measure Fisher Align is more stable across datasets (Spearman's $\rho$ of 0.368 on VLCS, 0.212 on PACS and 0.585 on RotatedMNIST). Our intuition for Fisher Align is that since the eigenvectors of the Fisher indicate the directions in parameter space in which the model is most sensitive, if these directions match on a target distribution, one is more likely to generalize.

### 5.3 Results with Compound Measures

In the previous sub-section we studied each of the proposed measures in isolation. Next, we study if combining the measures with in-domain test error, $\epsilon_S(\hat{c})$, can yield better performance. This is similar to the theory, which bounds the gap between target and source domain validation errors. In each case, we fit a regression model of the form $\alpha\epsilon_S(\hat{c}) + \beta * \text{feature}$ to predict $\epsilon_T(\hat{c})$ and then use the predictions to compute Spearman's $\rho$. However, we also report $R^2$ of the regression in table 1.

**Joint model across all environments.** We first perform the regression (as explained in section 3) across all datasets and environments jointly (table 1, "Joint" column). This does not allow the model to incorporate the specific environment in the weights of the regressor, but instead insists on the same predictor or "rule" across all environments. This allows a direct comparison to the experiments we discussed in the previous sub-section. We find that in this setup, only Entropy-Target, Jacobian and MMD-Mean-Cov when used in conjunction with $\epsilon_S(\hat{c})$ improve Spearman's $\rho$ (table 1), indicating that these measures might be complementary to $\epsilon_S(\hat{c})$.

**Stratified models for each environment.** We next allow the regressor to choose a new set of parameters for each environment. In such a setting, it is impossible to perform worse than $\epsilon_S(\hat{c})$, since one can always recover a solution which places 0 weight on the second feature if that is indeed the best solution to be found. As expected, table 1 reveals that all measures improve in conjunction with $\epsilon_S(\hat{c})$ in this setting. However, it is still instructive to study which measures improve the performance of $\epsilon_S(\hat{c})$ the most, relative to others. We find that MMD-Gaussian, MMD-Mean-Cov, Jacobian, Entropy-Target all lead to Spearman's $\rho$ values greater than 0.80 in this setting, suggesting them to be promising candidates to pursue further in this context.

**Which is better, $L_2$-Path Norm or Mixup?** While $L_2$-Path Norm appears to have a higher Spearman's $\rho$ than Mixup in the Alone setting (table 1), it appears that most of the benefit from $L_2$-Path Norm comes from it being a good predictor of in-distribution generalization (Neyshabur et al., 2015). When paired with $\epsilon_S(\hat{c})$, even in the stratified setting, we find that $L_2$-Path Norm has the same performance as $\epsilon_S(\hat{c})$ (0.712 Spearman's $\rho$), whereas Mixup improves $\epsilon_S(\hat{c})$ from 0.712 to 0.749 Spearman's $\rho$. Similar trends can be observed for $R^2$ as well.

## 6 Related work and Limitations

Our work first sets out to test the seminal theory around unsupervised domain adaptation (Ben-David et al., 2007, 2010), which aims to understand when an approach is expected to generalize to test domains different from the training domain. We provide an empirical evaluation of some of the bounds and theory, and surprisingly find that less principled approaches based on the Fisher information and Jacobian Norm perform better than bounds inspired by domain adaptation theory.

Most related to our current work is the work on DomainBed (Gulrajani & Lopez-Paz, 2020) which observed that more careful experimental choices lead to ERMs that are competitive with state of the art domain generalization methods. Our work seeks to understand what characterizes ERMs that generalize, with our results suggesting the entropy of the output distribution, norm of the Jacobian, and measures based on the Fisher being promising candidates for future research.

Other related work (Zhao et al., 2019; Wu et al., 2019) finds that the theory of unsupervised domain adaptation (Ben-David et al., 2007, 2010) when used to minimize domain divergences (Ganin et al., 2015) is not always useful in practice, in terms of reducing target domain error. Our work is different in that we study domain generalization (as opposed to adaptation) where one cannot minimize the

domain divergences explicitly. Correspondingly, we uncover that the theory is not useful in practice for predicting generalization as opposed to enforcing it.

Our methodological approach also takes inspiration from Jiang et al. (2019); Neyshabur et al. (2017) which aims to understand how overparameterized deep networks are able to achieve good generalization, motivated in part by their ability to fit random labels (Zhang et al., 2016). We implement some of the best performing measures from the large-scale empirical study of Jiang et al. (2019) for the in-domain setting, and employ rank correlation as a measure of performance following this work. We believe it is a fruitful intellectual and practical challenge to understand the role of implicit regularization (Neyshabur et al., 2015) in the domain generalization setting as well.

Previous work (Hoffman et al., 2019; Drucker & Le Cun, 1992; Varga et al., 2017) has studied domain generalization using the Jacobian norm as a regularizer. While their aim is to build a more robust model for supervised learning, our goal is in characterizing if an ERM trained without any regularization can be shown to be implicitly learning smooth functions, and therefore help aid understanding of the implicit domain generalization phenomenon. Novak et al. (2018) show that the Jacobian norm is useful for predicting generalization at the level of individual test points for in-domain classification. Here we extend their experimental setting to the out-of-distribution case, and show that the Jacobian is useful at the population level. Jastrzebski et al. (2020) recently studied the role of the evolution of the trace of the Fisher information over the course of training and its impact on in-domain generalization and proposed a regularizer based on this insight. We utilize the Fisher for the domain generalization setting. In contrast to Jastrzebski et al. (2020) who study the trace of the Fisher, our best Fisher-based method uses the eigenvectors for training and test.

Finally, predicting out-of-distribution generalization is related to out-of-distribution detection, but there is an important difference. While in out-of-distribution detection the task is to flag out-of-distribution examples that do not belong to any of the classes observed during training (Hendrycks et al., 2018), in out-of-distribution generalization we assume that test instances always belong to one of the classes of interest, albeit they are generated from a different distribution during test time.

**Limitations:** Our work is a step towards building an understanding of when networks trained with ERM generalize to novel target domains. However, our study is empirical, which limits the generality of our conclusions. We hope this effort spurs future work on theoretical aspects of this problem. While we follow the model architectures of Gulrajani & Lopez-Paz (2020) to obtain a large set of pretrained models, studying the impact of architectures in addition to hyperparameters is also an interesting line of study.

## 7    Conclusion

In this work, we studied the domain generalization performance of ERM trained deep neural networks. Surprisingly, we found that adapting theory from unsupervised domain adaptation was of limited use in predicting how well models generalize out-of-distribution. We found that the assumptions of the theory were satisfied, in particular $\lambda$-closeness, yet the practically accessible components of the theory do not provide strong predictions about which models will generalize. In an effort to discover other correlates of domain generalization, we studied the performance of a suite of novel empirical measures, finding a number of promising candidates. Our work provides a critical first step for understanding the implicit domain generalization properties of ERMs and points towards promising future theoretical investigations into strongly predictive measures of domain generalization.

## Social Impact

While most work that improves the performance of machine learning models can be leveraged by applications that have negative societal impact, we believe that the results of this paper actually take steps in a positive direction for the deployment of machine learning models in broader society. In particular, our work aims to understand when models that were not trained to be robust to domain shift break down or actually succeed on novel test domains. A better understanding and ability to predict model behavior in such situations should help in building more trustworthy machine learning methods that are less likely to result in unintended consequences when deployed on novel data. We emphasize that our work does not propose a new algorithm for domain generalization but rather attempts to *predict* when previously trained models can be trusted to perform in novel domains.

## Acknowledgments

We would like to thank Kamalika Chaudhuri, Pascal Vincent, Ricardo Monti, Edward Grefenstette, Yann Dubois and numerous other colleagues at Facebook for discussions and feedback on the paper. We also thank Kamalika for detailed feedback on a draft of the paper.

## Funding Transparency Statement

D.J.S was partially supported by the NIH under award R01EB026943, the NSF through the Center for the Physics of Biological Function (PHY-1734030), and as a Simons Investigator in the Mathematical Modeling of Living Systems (MMLS).

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
