# An Empirical Investigation of Domain Generalization with Empirical Risk Minimizers
# (Appendix)

## 1. Clarification on Notation

To be consistent with the theory, we denoted the test error and source error as $\epsilon_T(\hat{h})$ and $\epsilon_S(\hat{h})$ in the main paper, but this obscures the fact that the theory derives bounds given a representation $\mathcal{R}$ (Ben-David et al., 2010). In order to make the detailed exposition of measures more clear, for the rest of this document, we instead refer to the empirical risk minimizer as $\hat{c}$, and the source and target errors as $\epsilon_S(\hat{c})$ and $\epsilon_T(\hat{c})$ respectively, and use $h$ to refer only to mappings from the representation space $\mathcal{R}$ to the label space $\mathcal{Y}$.

## 2. Variance of Estimates of $\mathcal{H}$-divergence and $\mathcal{H}\Delta\mathcal{H}$-divergence

In this section we detail the variance exhibited by the estimators for the $\mathcal{H}$-divergence and $\mathcal{H}\Delta\mathcal{H}$-divergence, which are core divergence measures studied by the theory from (Ben-David et al., 2010; 2007). For algorithmic details of how we estimate these measures, see section 7.

In order to estimate the variance, we pick a random subset of 10 models on VLCS and RotatedMNIST, and estimate the divergence measure of interest by bootstrapping with 80% of the original data. We then compute the mean $\mu$ and standard deviation $\sigma$ for the bootstrapped estimates, and report the Signal to Noise Ratio (SNR) as $\frac{\mu}{\sigma}$. We repeat this process across multiple models, and report the mean SNR for each of the estimators we use. See table 1 for the results.

We note that -divergence has the highest signal to noise ratio across both the datasets, whereas the $\mathcal{H}\Delta\mathcal{H}$-divergence based measures are estimated with a lower signal to noise ratio. These trends indicate that the bounds and measures estimated with $\mathcal{H}$-divergence might be more accurate and any good or bad performance is due to the measure being good or bad, as opposed to the $\mathcal{H}\Delta\mathcal{H}$-divergence, where there might be significant estimation error. Interestingly, we notice that across both datasets, the multi source (MS) versions have a higher SNR than single source (SS) versions.

.

Table 1: Different divergence measures (rows) and the Signal-to-Noise Ratio (SNR) (columns) on VLCS and RotatedMNIST. We notice that $\mathcal{H}\Delta\mathcal{H}$-divergence MS estimates have better SNR than $\mathcal{H}\Delta\mathcal{H}$-divergence SS but both of them are worse than $\mathcal{H}$-divergence, echoing the theoretical results from (Ben-David et al., 2010).

| Measure | SNR | |
|---|---|---|
| | VLCS | RotatedMNIST |
| $\mathcal{H}$-divergence | 22.59 | 22.20 |
| $\mathcal{H}\Delta\mathcal{H}$-divergence SS | 9.10 | 1.95 |
| $\mathcal{H}\Delta\mathcal{H}$-divergence MS | 15.48 | 3.10 |

## 3. Additional Results

**Comparison of the CORAL algorithm to ERM.** How does the choice of algorithm for domain generalization influence the trends discussed in the main paper? To obtain a sense for this question, we train 3000 models using the Deep CORAL (Sun & Saenko, 2016) approach on VLCS and repeat the analysis from the main paper on the models trained with CORAL. The Deep CORAL approach essentially matches the mean and variance of the intermediate layers across input domains, and thus offers a counterpoint to ERM which does not use any per-domain information. Further, to the best of our knowledge, it is the only approach to outperform ERM overall on DomainBed making it a good candidate for this analysis.

We find that the performance of most of the measures remains consistent across both the training algorithms, which is encouraging (fig. 1).

**Results with multiple variables for regression** We next perform regression in the Joint setting (Sec.5.3, main paper) where we fit a regression model across all environments, with 5 features instead of 2 reported in the main paper. We find that it is possible to get an Spearman's $\rho$ of 80.4 when using Entropy-Source, Entropy-Target, MMD-Gaussian, MMD-Mean-Cov, Sharpness, and $\epsilon_S(\hat{c})$ as fea-

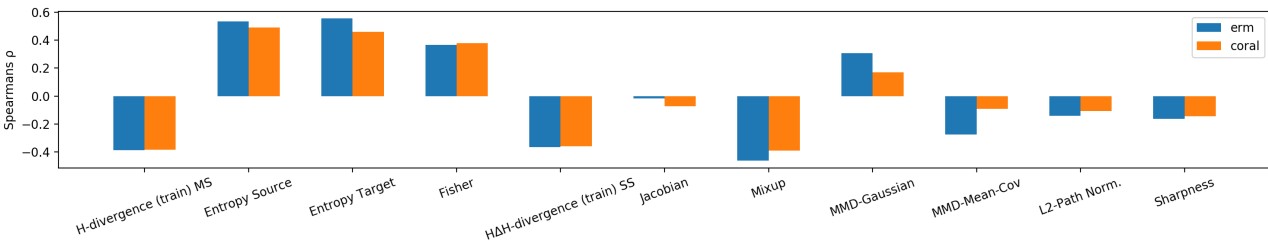

Figure 1: Spearman's $\rho$ (y-axis) plotted against measure (x-axis) for the ERM and CORAL algorithms for VLCS. We note that the performance of most of the measures remains consistent across ERM and CORAL, except MMD-Gaussian, MMD-Mean-Cov and Jacobian based measures.

tures for the regression.

**Scatter plots of different measures and observed test error $\epsilon_T(\hat{c})$.** We next present scatter plots of different measures with respect to the error on the test domain, to provide an intuitive sense of the data on which we perform our regressions that use one measure at a time to predict generalization (fig. 5-fig. 15, at the end of the document). In order to show these scatter plots, we drop all the datapoints for which a measure has a value greater than $1e3$, which clears all the outliers from the plots and allows better visualization. However, for quantitative analysis we retain all the datapoints.

## 4. Performance of All Measures Considered

We considered a set of 40 metrics overall and report only a small subset of them in the main paper. In table 2 we provide detailed results of all the measures we study. See section 7 for detailed algorithmic explanations of the implementations of each of the measures listed here. Also shown in fig. 2 is a distance matrix computed based on Spearman's $\rho$ visualizing the dependencies of the measures with each other.

## 5. Canonicalization of the Measures

fig. 3 provides details of the canonicalization performed on each of the measures as explained in the main paper. Interestingly, for measures such as the path norm, we find that the canonicalization in this setting is opposite of what is conventionally understood. In general, a high path norm should indicate higher test error, but here the opposite seems to be true.

## 6. Connection of DA-GM to theoretical results from **Ben-David et al. (2007)**

We were inspired by results from the theory and practice of domain adaptation to construct generalization measures in this category. In particular, (Ben-David et al., 2007) prove

bounds on the target domain performance that depend on the ability of a classifier to distinguish samples from the source and target domains. As described in the main text, they find that the target domain error can be bounded by the source domain error plus a term that is related to the sample $\mathcal{H}$-distance between the source and target domains with respect to the hypothesis class of the model plus the degree of $\lambda$-closeness of the hypothesis family. This $\mathcal{H}$-distance, $d_{\mathcal{H}}(\tilde{D}_S, \tilde{D}_T)$, in turn can be calculated by finding the optimal performance of a classifier trained to distinguish samples from the source and target domains. Here, $\tilde{D}_S$ ($\tilde{D}_T$) are the induced distribution of source (target) domains pushed to the representational space. (Note that in this section we use this notation to make contact with the existing theory, but in subsequent sections, these distributions are called $\mathcal{S}_{tr}^z$ and $\mathcal{T}_{tr}^z$, respectively.) We emphasize that these results assume a fixed representational space and that the entire classifier $c$ is composed of an encoding into this space followed by a decoder $h \in \mathcal{H}$ that takes encodings and predicts target labels. Specifically, the target domain test error is upper bounded by

$$\epsilon_T(c) \leq \epsilon_S(c) + d_{\mathcal{H}}(\tilde{D}_S, \tilde{D}_T) + \lambda, \qquad (1)$$

where $\lambda = \min_{h \in \mathcal{H}}(\epsilon_T(h) + \epsilon_S(h))$.

We also develop measures based on follow-up theoretical work in (Ben-David et al., 2010) on divergence measures using the symmetric difference hypothesis space. $\mathcal{H}\Delta\mathcal{H}$, which is defined as the set of hypothesis of the form $g = h(x) \oplus h'(x)$, where $h, h' \in \mathcal{H}$ and $\oplus$ is the XOR function. That is, the symmetric difference hypothesis space is the set of all disagreements between hypotheses in our hypothesis class. This object is important for the theory to bound the target domain test error in multiple settings, i.e. when multiple different environments are used for training. Here we summarize a result from (Ben-David et al., 2010), similar to a specialization of their Theorem 4, which focuses on multi-source training. (Ben-David et al., 2010) proves that if we train an ERM, $\hat{c}$, on (equally-weighted) source domains $j = 1...N$, resulting in an (equally-weighted) source

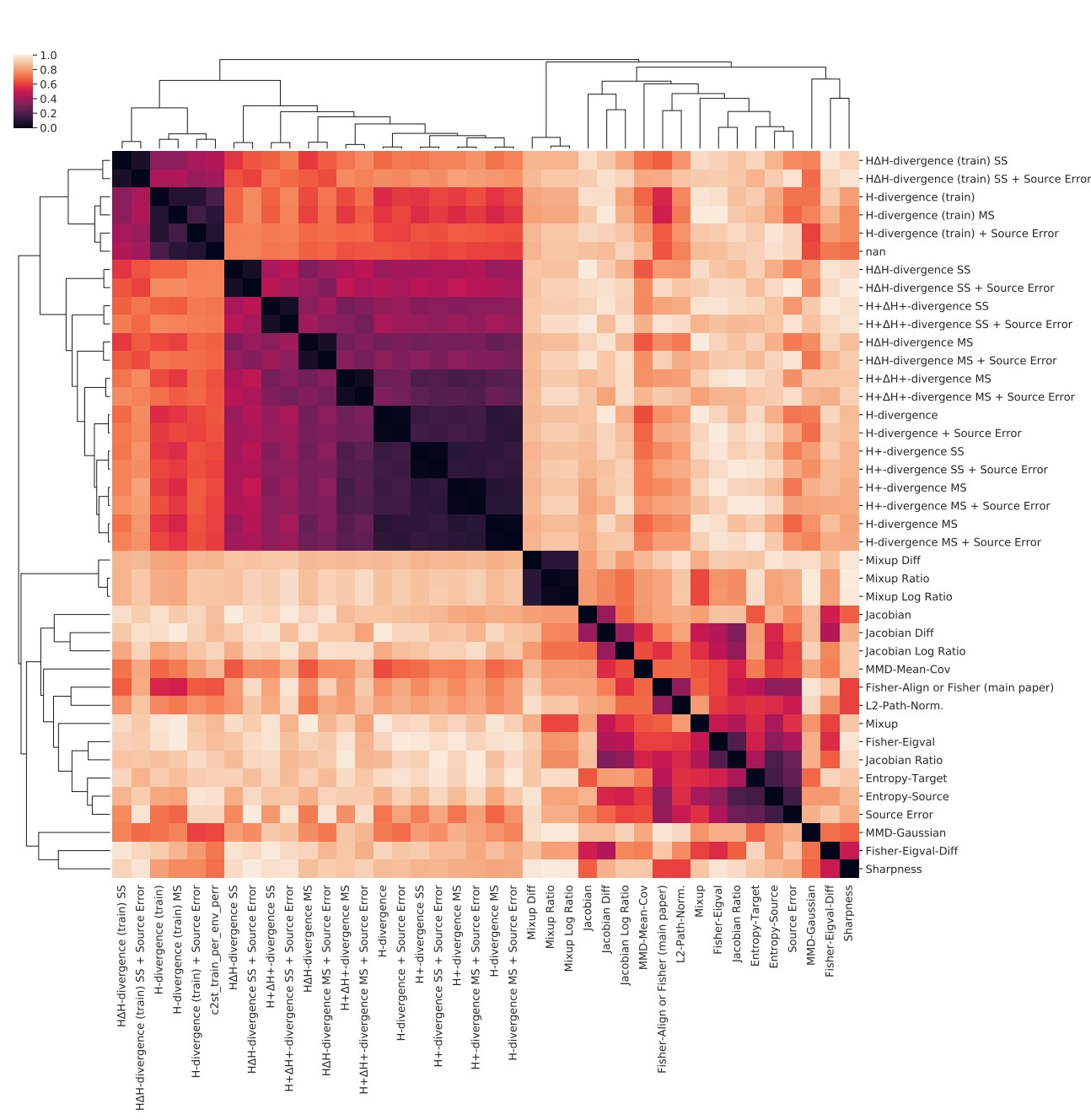

Figure 2: Distances (1-Spearman's $\rho$) matrix plotting the relationship between the $i^{th}$ row and $j^{th}$ column. Rows and columns are organized and grouped on the basis of hierarchical clustering of the measures based on similarity.

Table 2: Measure (rows) against Spearman's $\rho$ (column) for all the measures considered in the analysis for the paper. The main paper contains results with a subset of promising results, whereas here we provide results for a more exhaustive set of measures. See section 7 for more details on the meaures.

| measure | Spearman's $\rho$ |
| --- | --- |
| Entropy-Target | 0.733858 |
| Source Error | 0.711895 |
| Entropy-Source | 0.598143 |
| Fisher-Align or Fisher (main paper) | 0.441314 |
| Jacobian | 0.419960 |
| Jacobian Ratio | 0.397433 |
| L2-Path-Norm. | 0.389819 |
| Fisher-Eigval | 0.298458 |
| MMD-Gaussian | 0.283357 |
| Mixup | 0.278850 |
| $\mathcal{H}\Delta\mathcal{H}$-divergence (train) SS + Source Error | 0.154016 |
| $\mathcal{H}\Delta\mathcal{H}$-divergence SS + Source Error | 0.152128 |
| $\mathcal{H}$-divergence (train) MS | 0.151395 |
| $\mathcal{H}\Delta\mathcal{H}$-divergence MS + Source Error | 0.136513 |
| $\mathcal{H}^+\Delta\mathcal{H}^+$-divergence SS + Source Error | 0.124091 |
| Sharpness | 0.117217 |
| $\mathcal{H}^+$-divergence MS | 0.113737 |
| $\mathcal{H}$-divergence (train) | 0.106712 |
| $\mathcal{H}$-divergence MS | 0.105065 |
| MMD-Mean-Cov | 0.099831 |
| $\mathcal{H}^+\Delta\mathcal{H}^+$-divergence MS | 0.079718 |
| Fisher-Eigval-Diff | 0.078561 |
| Jacobian Log Ratio | 0.068404 |
| $\mathcal{H}^+$-divergence SS | 0.064583 |
| $\mathcal{H}^+\Delta\mathcal{H}^+$-divergence MS + Source Error | 0.063797 |
| Mixup Diff | 0.061810 |
| $\mathcal{H}\Delta\mathcal{H}$-divergence MS | 0.057434 |
| $\mathcal{H}$-divergence (train) + Source Error | 0.057375 |
| $\mathcal{H}^+\Delta\mathcal{H}^+$-divergence SS | 0.055414 |
| $\mathcal{H}$-divergence | 0.049011 |
| $\mathcal{H}\Delta\mathcal{H}$-divergence (train) SS | 0.045932 |
| $\mathcal{H}^+$-divergence MS + Source Error | 0.042193 |
| $\mathcal{H}\Delta\mathcal{H}$-divergence SS | 0.038274 |
| $\mathcal{H}$-divergence MS + Source Error | 0.029553 |
| $\mathcal{H}$-divergence + Source Error | 0.027533 |
| Jacobian Diff | 0.018869 |
| Mixup Ratio | 0.017812 |
| Mixup Log Ratio | 0.017766 |
| $\mathcal{H}^+$-divergence SS + Source Error | 0.006798 |

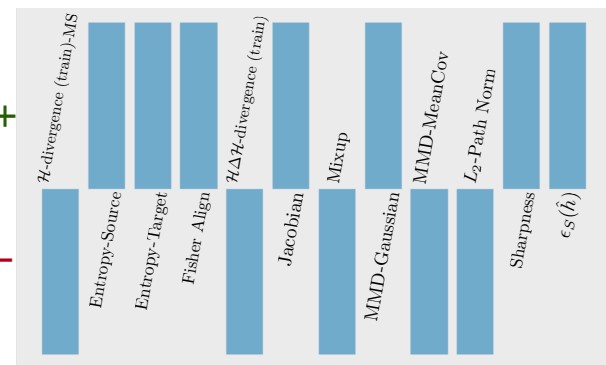

Figure 3: Sign multiplied to the measure (+1 or -1) to canonicalize the measure to have a positive correlation with $\epsilon_T(\hat{c})$. This canonicalization is used to report the results in Sec. 5 of the main paper.

domain test error $\epsilon_S(\hat{c})$, then the target domain test error of $\hat{c}$ can be bounded by

$$\epsilon_T(\hat{c}) \leq \epsilon_S(\hat{c}) + \frac{1}{N}\sum_{j=1}^{N}\left(\lambda_j + \frac{1}{2}d_{\mathcal{H}\Delta\mathcal{H}}(\tilde{D}_j, \tilde{D}_T)\right). \quad (2)$$

Here, $\lambda_j = \min_{h\in\mathcal{H}}(\epsilon_T(h) + \epsilon_j(h))$ is the degree of $\lambda$-closeness between the source domain $j$ and the target domain $T$ and $\tilde{D}_j$ is the distribution of source domain $j$ pushed to the representational space.

## 7. Algorithmic Details

In this section we provide more details on some of the generalization measures we compute in the main paper along with a more comprehensive set of measures we study, and provide algorithm-level details. Before diving into the details of the measures, we first explain notation (which is slightly different from the main paper, but more specific and detailed to enable a more precise characterization of what is done in the various generalization measures).

### 7.1. Notation

Given $N$ examples $\mathcal{S}_{\text{tr}} = \{(X_i, y_i)\}_{i=1}^{N} \sim p(y|X)p_{\text{train}}(X)$, a trained ERM $\hat{c} : \mathcal{X} \to \mathcal{P}(\mathcal{Y})$ which computes probabilistic predictions in the label space $\mathcal{Y}$, and $M$ examples of held out data $\mathcal{T}_{\text{tr}} = \{X_i'\}_{i=1}^{M} \sim p_{\text{test}}(X)$, a generalization measure for domain generalization aims to predict how well $\hat{c}$ will generalize to $\mathcal{T}_{\text{te}} = \{X_i^\dagger, y_i^\dagger\} \sim p(y|X)p_{\text{test}}(X)$. We will often decompose the function as $c(\cdot) = h(e(\cdot))$ where $e$ is an encoder $\mathcal{X} \to \mathcal{Z}$, $z \in \mathcal{Z}$ is a representation, and $h : \mathcal{Z} \to \mathcal{Y}$ is a classifier in a hypothesis class . Unless stated otherwise we set $z$ as the last layer of the network.

## 7.2. Measures based on theory

$\mathcal{H}$-**divergence:** We explain in more detail how we compute the classifier two sample test measure below. Most of the explanation is derived from (Lopez-Paz & Oquab, 2016), which we reproduce in our notation for convenience. Given an encoder $\mathcal{R}: \mathcal{X} \to \mathcal{Z}$, which encodes inputs $X$ into a representation $Z$, a chosen function family for classification $\mathcal{H}: \mathcal{Z} \to \mathcal{P}(\mathcal{Y})$, we follow the steps in algorithm 1.

---

**Algorithm 1** Computing $\mathcal{H}$-divergence measure

1: **Given:** encoder $e$, $\mathcal{S}_{\text{tr}}$, $\mathcal{T}_{\text{tr}}$, classifier family $\mathcal{H}$
2: compute $\mathcal{S}_{\text{tr}}^z = \{e(X) \,\forall X, y \in \mathcal{S}_{\text{tr}}\}$ and $\mathcal{T}_{\text{tr}}^z = \{e(X) \,\forall X, y \in \mathcal{T}_{\text{tr}}\}$
3: ensure that $\mathcal{S}_{\text{tr}}$ and $\mathcal{T}_{\text{tr}}$ have the same number of datapoints, drop any additional / extra datapoints at random if there are more datapoints in either of the sets
4: Given an indicator function for the set $\mathbf{I}$, construct a dataset $\mathcal{U} = \{Z, I\,[Z \in \mathcal{S}_{\text{tr}}] : \forall X \in \mathcal{S}_{\text{tr}}\} \bigcup \{Z, I\,[Z \in \mathcal{S}_{\text{tr}}] : \forall X \in \mathcal{T}_{\text{tr}}\}$.
5: Split $\mathcal{U}$ into two disjoint sets (deterministically) $\mathcal{U}_{\text{tr}}, \mathcal{U}_{\text{te}}$ such that $\mathcal{U}_{\text{tr}} \bigcup \mathcal{U}_{\text{te}} = \mathcal{U}$
6: Fit $\mathcal{H}$ on $\mathcal{U}_{\text{tr}}$ using log-loss $\min_{h \in \mathcal{H}} \sum_{Z,y \in \mathcal{U}_{\text{tr}}} \log h(Z)[y]$
7: **Return:** $\sum_{Z,y \in \mathcal{U}_{\text{te}}} \mathbf{I}\,[\text{argmax}(h(Z)) = y]$

---

$\mathcal{H}^+$-**divergence:** We get the C2ST-Big measure when we set $\mathcal{H} = \mathcal{H}^+$ in the above algorithm to be a larger function class ($\mathcal{H}^+$) than $\mathcal{H}^\dagger$ used in the ERM $\hat{c}$, where $\hat{c}(X) = h^+(e(X))$ and $h^+ \in \mathcal{H}^\dagger$. In practice our original function family $\mathcal{H}^\dagger$ is set to the linear function family, while the larger function family $\mathcal{H}^+$ is set to an MLP with layer sizes (num features/2, num features/4, num features/4) and ReLU nonlinearity.

$\mathcal{H}\Delta\mathcal{H}$-**divergence:** As explained in the main paper, this divergence measure was proposed in (Ben-David et al., 2010). Denoting by $\mathcal{S}^z$ the featurized version of the source data and $\mathcal{T}^z$ the featurized version of the test data, $\mathbf{I}$ the indicator function, the $\mathcal{H}\Delta\mathcal{H}$-divergence is defined as follows:

$$d_{\mathcal{H}\Delta\mathcal{H}} = \min_{h,h' \in \mathcal{H}} E_{\mathcal{S}^z}\mathbf{I}[h(z) \neq h'(z)] - E_{\mathcal{T}^z}\mathbf{I}[h(z) = h'(z)] \tag{3}$$

Intuitively, this means that two domains are more different if one can find two members $h, h' \in \mathcal{H}$ such that they maximally disagree on the source domain and maximally agree on the test domain.

We estimate this divergence measure by training two networks $h, h'$ as shown in algorithm 2.

$\mathcal{H}^+\Delta\mathcal{H}^+$-**divergence:** Similar to $\mathcal{H}$-divergence, when we use a larger function family $\mathcal{H}^+$ we obtain a version of $\mathcal{H}\Delta\mathcal{H}$ divergence called $\mathcal{H}^+\Delta\mathcal{H}^+$-divergence.

---

**Algorithm 2** Computing $\mathcal{H}\Delta\mathcal{H}$-divergence measure
.

1: **Given:** encoder $e$, $\mathcal{S}_{\text{tr}}$, $\mathcal{T}_{\text{tr}}$, classifier family $\mathcal{H}$, $rand(\mathcal{Y})$, a random label generator in the space $\mathcal{Y}$
2: compute $\mathcal{S}_{\text{tr}}^z = \{e(X) \,\forall X, y \in \mathcal{S}_{\text{tr}}\}$ and $\mathcal{T}_{\text{tr}}^z = \{e(X) \,\forall X, y \in \mathcal{T}_{\text{tr}}\}$
3: ensure that $\mathcal{S}_{\text{tr}}$ and $\mathcal{T}_{\text{tr}}$ have the same number of datapoints, drop any additional / extra datapoints at random if there are more datapoints in either of the sets
4: Construct a dataset $\mathcal{U} = \{z, rand(\mathcal{Y}) : z \in \mathcal{S}_{\text{tr}}^z\} \bigcup \{z, \text{argmax}(h'(z)) : z \in \mathcal{T}_{\text{tr}}^z\}$ and $\mathcal{U}' = \{z, rand(\mathcal{Y}) : z \in \mathcal{S}_{\text{tr}}^z\} \bigcup \{z, \text{argmax}(h(z)) : z \in \mathcal{T}_{\text{tr}}^z\}$.
5: Split $\mathcal{U}$ into two disjoint sets (deterministically) $\mathcal{U}_{\text{tr}}, \mathcal{U}_{\text{te}}$ such that $\mathcal{U}_{\text{tr}} \bigcup \mathcal{U}_{\text{te}} = \mathcal{U}$
6: Split $\mathcal{U}'$ into two disjoint sets (deterministically) $\mathcal{U}'_{\text{tr}}, \mathcal{U}'_{\text{te}}$ such that $\mathcal{U}'_{\text{tr}} \bigcup \mathcal{U}'_{\text{te}} = \mathcal{U}'$
7: Take a gradient step updating $h$ on $\mathcal{U}_{\text{tr}}$ using log-loss $\min_{h \in \mathcal{H}} \sum_{Z,y \in \mathcal{U}_{\text{tr}}} \log h(Z)[y]$ and $h'$ on $\mathcal{U}'$ using log-loss $\min_{h' \in \mathcal{H}} \sum_{Z,y \in \mathcal{U}'_{\text{tr}}} \log h(Z)[y]$
8: Repeat from Step 4 until convergence
9: **Return:** $\frac{1}{|\mathcal{U}_{te}|} \sum_{z,y \in \mathcal{U}_{te}} \mathbf{I}[h(z) = y] + \frac{1}{|\mathcal{U}'_{te}|} \sum_{z,y \in \mathcal{U}'_{te}} \mathbf{I}[h'(z) = y]$

---

$\mathcal{H}$-**divergence (train):** We use algorithm 3 for $\mathcal{H}$-divergence (train), with a modification to the last line of the algorithm for $\mathcal{H}$-divergence.

---

**Algorithm 3** Computing $\mathcal{H}$-divergence (train) Measure

1: **Given:** encoder $e$, $\mathcal{S}_{\text{tr}}$, $\mathcal{T}_{\text{tr}}$, classifier family $\mathcal{H}$
2: compute $\mathcal{S}_{\text{tr}}^z = \{e(X) \,\forall X, y \in \mathcal{S}_{\text{tr}}\}$ and $\mathcal{T}_{\text{tr}}^z = \{e(X) \,\forall X, y \in \mathcal{T}_{\text{tr}}\}$
3: ensure that $\mathcal{S}_{\text{tr}}$ and $\mathcal{T}_{\text{tr}}$ have the same number of datapoints, drop any additional / extra datapoints at random if there are more datapoints in either of the sets
4: Given an indicator function for the set $\mathbf{I}$, construct a dataset $\mathcal{U} = \{Z, I\,[Z \in \mathcal{S}_{\text{tr}}] : \forall X \in \mathcal{S}_{\text{tr}}\} \bigcup \{Z, I\,[Z \in \mathcal{S}_{\text{tr}}] : \forall X \in \mathcal{T}_{\text{tr}}\}$.
5: Split $\mathcal{U}$ into two disjoint sets (deterministically) $\mathcal{U}_{\text{tr}}, \mathcal{U}_{\text{te}}$ such that $\mathcal{U}_{\text{tr}} \bigcup \mathcal{U}_{\text{te}} = \mathcal{U}$
6: Fit $\mathcal{H}$ on $\mathcal{U}_{\text{tr}}$ using log-loss $\min_{c \in \mathcal{H}} \sum_{Z,y \in \mathcal{U}_{\text{tr}}} \log c(Z)[y]$
7: **Return:** $\sum_{Z,y \in \mathcal{U}_{\text{tr}}} \mathbf{I}\,[\text{argmax}(c(Z)) = y]$

---

$\mathcal{H}\Delta\mathcal{H}$-**divergence (train)** We use a nearly identical algorithm (algorithm 4) to that for $\mathcal{H}\Delta\mathcal{H}$-divergence, with change to the last line (in blue).

**Multi-Source (MS) versus. Single-Source (SS).** In the presence of multiple source environments, we usually treat all the source environments as one environment and compute the measures as explained above. However, based on the theory from (Ben-David et al., 2010) we also consider

**Algorithm 4** Computing $\mathcal{H}\Delta\mathcal{H}$-divergence (train) measure

1: **Given:** encoder $e$, $\mathcal{S}_{\text{tr}}$, $\mathcal{T}_{\text{tr}}$, classifier family $\mathcal{H}$, $rand(\mathcal{Y})$, a random label generator in the space $\mathcal{Y}$
2: compute $\mathcal{S}_{\text{tr}}^z = \{e(X) \,\forall X, y \in \mathcal{S}_{\text{tr}}\}$ and $\mathcal{T}_{\text{tr}}^z = \{e(X) \,\forall X, y \in \mathcal{T}_{\text{tr}}\}$
3: ensure that $\mathcal{S}_{\text{tr}}$ and $\mathcal{T}_{\text{tr}}$ have the same number of datapoints, drop any additional / extra datapoints at random if there are more datapoints in either of the sets
4: Construct a dataset $\mathcal{U} = \{z, rand(\mathcal{Y}) : z \in \mathcal{S}_{\text{tr}}^z\} \bigcup \{z, \text{argmax}(h'(z)) : z \in \mathcal{T}_{\text{tr}}^z\}$ and $\mathcal{U}' = \{z, rand(\mathcal{Y}) : z \in \mathcal{S}_{\text{tr}}^z\} \bigcup \{z, \text{argmax}(h(z)) : z \in \mathcal{T}_{\text{tr}}^z\}$.
5: Split $\mathcal{U}$ into two disjoint sets (deterministically) $\mathcal{U}_{\text{tr}}, \mathcal{U}_{\text{te}}$ such that $\mathcal{U}_{\text{tr}} \bigcup \mathcal{U}_{\text{te}} = \mathcal{U}$
6: Split $\mathcal{U}'$ into two disjoint sets (deterministically) $\mathcal{U}'_{\text{tr}}, \mathcal{U}'_{\text{te}}$ such that $\mathcal{U}'_{\text{tr}} \bigcup \mathcal{U}'_{\text{te}} = \mathcal{U}'$
7: Take a gradient step updating $h$ on $\mathcal{U}_{\text{tr}}$ using log-loss $\min_{h \in \mathcal{H}} \sum_{Z, y \in \mathcal{U}_{\text{tr}}} \log h(Z)[y]$ and $h'$ on $\mathcal{U}'$ using log-loss $\min_{h' \in \mathcal{H}} \sum_{Z, y \in \mathcal{U}'_{\text{tr}}} \log h(Z)[y]$
8: Repeat from Step 4 until convergence
9: **Return:** $\frac{1}{|\mathcal{U}_{tr}|} \sum_{z, y \in \mathcal{U}_{te}} \mathbf{I}[h(z) = y] + \frac{1}{|\mathcal{U}_{tr}|} \sum_{z, y \in \mathcal{U}'_{te}} \mathbf{I}[h'(z) = y]$

---

measures which study things at the level of multiple sources. To do this, we follow the theory section 6 and compute the divergence measure between each source domain and target domain, and report the overall measure as the mean of the divergence measures for each source domain, taken in turn, relative to a given target domain. Measures computed in this fashion are suffixed with Multi Source (MS). By default, all measures are computed and reported in a Single Source (SS) manner.

**Divergence + Source Error.** Finally, we also compute measures of generalization which take one of the theory inspired divergences above and consider the sum of the divergence and the error on the source domain $\epsilon_S(\hat{c})$ as the measure, in line with the bounds from theory section 6.

### 7.3. Other Empirical Measures

$L_2$-**Path Norm:** The path norm is computed following the procedure from (Jiang et al., 2019). Let the ERM $\hat{c}_\theta$ be parameterized by $\theta \in R^K$, where $K$ is the number of parameters of the model. One can then compute the path norm by squaring the parameters of the network, passing an all 1 input through the network and computing the L2 norm of the logits from the network. That is, given inputs $X \in R^D$, $\hat{c}(\cdot) = \text{softmax}(g^*(\cdot))$, we compute path norm as follows:

**Sharpness:** We compute the sharpness bound using Algorithm 3 in Apppendix D from Jiang et al. (2019), with the suggested changes for magnitude aware perturbation.

**Algorithm 5** $L_2$-Path Norm

1: **Given:** $g_\theta^*$, $D$
2: $\theta \leftarrow \theta^2$
3: $X_1 \leftarrow \mathbf{1} \in R^D$
4: $Y \leftarrow g_\theta^*(X_1)$
5: **Return:** sqrt $\left( \sum_{y_i \in Y} y_i \right)$

---

Note that there is an errata in the heading of Algorithm 3, it should be $\alpha$ not $\sigma$.

**MMD-Gaussian and MMD-Mean-Cov:** Similar to $\mathcal{H}$-divergence measures, the MMD measure also works in the representation space, $\mathcal{Z}$, where given $\mathcal{S}_{\text{tr}}^z = \{e(X) \; X_i, y_i \in \mathcal{S}_{\text{tr}}\}$ and $\mathcal{T}_{\text{tr}}^z = \{e(X) \,\forall X, y \in \mathcal{T}_{\text{tr}}\}$, we use a kernel based measure to compute the similarity between the domains. Since these measures do not use classification, we do not drop any datapoints which are extra in the source or the target domains, unlike Step 3 in algorithm 3. Accordingly, let $M$ be the number of datapoints in $\mathcal{S}_{\text{tr}}^z$ and $N$ be the number of datapoints in $\mathcal{T}_{\text{tr}}^z$.

**MMD-Gaussian:** We compute three kernels matrices, namely $K(\mathcal{S}_{\text{tr}}^z, \mathcal{S}_{\text{tr}}^z) \in R^{M \times M}$, $K(\mathcal{T}_{\text{tr}}^z, \mathcal{T}_{\text{tr}}^z) \in R^{N \times N}$, and $K(\mathcal{S}_{\text{tr}}^z, \mathcal{T}_{\text{tr}}^z) \in R^{M \times N}$. The MMD-Gaussian measure is then given by:

$$\text{MMD-Gaussian} = \frac{1}{M^2} \sum_{i=1}^{M} \sum_{j=1}^{M} K(\mathcal{S}_{\text{tr}}^z, \mathcal{S}_{\text{tr}}^z)_{ij} +$$
$$\frac{1}{N^2} \sum_{i=1}^{N} \sum_{j=1}^{N} K(\mathcal{T}_{\text{tr}}^z, \mathcal{T}_{\text{tr}}^z)_{ij} -$$
$$\frac{2}{M \times N} \sum_{i=1}^{M} \sum_{j=1}^{N} K(\mathcal{S}_{\text{tr}}^z, \mathcal{T}_{\text{tr}}^z)_{ij} \quad (4)$$

A kernel function $k$ is used to compute each of the entries in the kernel matrices above. For example, given $Z_i \in \mathcal{S}_{\text{tr}}^z$, the $i^{th}$ datapoint in the set and $Z_j \in \mathcal{T}_{\text{tr}}^z$, the $j^{th}$ datapoint in the set, the following are equivalent:

$$K(\mathcal{S}_{\text{tr}}, \mathcal{T}_{\text{tr}})_{ij} = k(Z_i, Z_j)$$

MMD-Gaussian uses a sum of Radial Basis Function (RBF) kernel functions, where given $\gamma \in \mathcal{G}$, such that $\mathcal{G} : [0.001, 0.01, 0.1, 1, 1, 10, 100, 1000]$ the final kernel can be expressed as:

$$k(Z_1, Z_2) = \sum_{\gamma \in \mathcal{G}} k_\gamma(Z_1, Z_2) \quad (5)$$
$$= \sum_{\gamma \in \mathcal{G}} \exp(-\gamma ||Z_1 - Z_2||_2^2) \quad (6)$$

**MMD-Mean-Cov:** For ease of explanation below we drop the subscript tr and the superscript $z$ and leave it understood that we are always working in the training splits of the source and target domains, and that we are working in the representation space $\mathcal{Z}$. Thus we write $\mathcal{S}_{\text{tr}}^z$ as $\mathcal{S}$ and $\mathcal{T}_{\text{tr}}^z$ as $\mathcal{T}$. With this notational simplification, given $\mathcal{U} \in \{\mathcal{S}, \mathcal{T}\}$, we first compute the sample mean:

$$\mu_{\mathcal{U}} = \frac{1}{|\mathcal{U}|} \sum_{Z_i \in \mathcal{U}} Z_i \tag{7}$$

Next, $Z_i \in R^K$, then we compute the sample covariance matrix $\text{cov} \in R^{K \times K}$, where the $i^{\text{th}}$ row and $j^{\text{th}}$ column are given by:

$$\text{cov}_{\mathcal{U}ij} = \frac{1}{|\mathcal{U}| - 1} \sum_{t=1}^{|\mathcal{U}|} (Z_{ti} - \mu_{\mathcal{U}i})(Z_{tj} - \mu_{\mathcal{U}j}) \tag{8}$$

Denoting by $\mathcal{F}$, the Frobenious norm of a matrix, the final measure is expressed as:

$$\text{MMD-Mean-Cov} = ||\mu_{\mathcal{S}} - \mu_{\mathcal{T}}||_2^2 + ||\text{cov}_{\mathcal{S}} - \text{cov}_{\mathcal{T}}||_{\mathcal{F}}^2 \tag{9}$$

**Fisher-based measures:** Given $\mathcal{U} \in \{\mathcal{S}_{\text{tr}}, \mathcal{T}_{\text{tr}}\}$, as explained in the main paper we compute the approximate Fisher information matrix using $\tilde{N}$ examples (see section 8 below for choices of $\tilde{N}$ for different datasets). We then perform an eigendecomposition to obtain the top $\tilde{N}$ approximate eigenvalues $\alpha_{\mathcal{U}} = \{\alpha_{\mathcal{U}}^1, \cdots, \alpha_{\mathcal{U}}^{\tilde{N}}\}$ and their corresponding eigenvectors $\mathcal{V}_{\mathcal{U}} = \{V_{\mathcal{U}}^1, \cdots, V_{\mathcal{U}}^{\tilde{N}}\}$. Given these we compute the following measures, as explained in the main paper:

1. Fisher-Eigval-Diff: Computes a measure $\sum_{n=1}^{\tilde{N}} (\alpha_{\mathcal{T}}^n - \alpha_{\mathcal{S}}^n)$

2. Fisher-Eigval-Ratio: Computes a measure $\frac{\sum_{n=1}^{\tilde{N}} \alpha_{\mathcal{T}}^n}{\sum_{n=1}^{\tilde{N}} \alpha_{\mathcal{S}}^n}$

3. Fisher Align: Computes the best match between the sets of eigenvectors $\mathcal{V}_{\mathcal{T}}$ and $\mathcal{V}_{\mathcal{S}}$ using the Hungarian algorithm (Kuhn, 1955) and reports the score of the best alignment between the sets, as the similarity between them. The similarity is defined by the cosine similarity matrix which is computed between unit vectors in the direction of the eigenvectors. Concretely, given eigenvectors $V_{\mathcal{S}}^i$ and $V_{\mathcal{T}}^j$, the corresponding row and column of a similarity matrix $\text{sim} \in R^{\tilde{N} \times \tilde{N}}$ can be computed as: $\text{sim}_{ij} = \frac{V_{\mathcal{S}}^{iT} \cdot V_{\mathcal{T}}^j}{||V_{\mathcal{S}}^i||_2 \cdot ||V_{\mathcal{T}}^j||_2}$. This similarity matrix is then fed to the Hungarian algorithm which returns the maximum similarity.

**Jacobian Norm Based Measures.** The input-output Jacobian, a measure of sensitivity of the model output to changes in the input, is mathematically defined as $J_{i,\alpha} = \frac{\partial h_i(x)}{\partial x_\alpha}$. Intuitively, a model that is less sensitive to changes in the input will generalize better. Thus, it has been studied in the context of robust learning (Hoffman et al., 2019) and in-distribution generalization (Novak et al., 2018), where it was found to be predictive of generalization at the level of individual test points. Similar to (Novak et al., 2018), we compute the Frobenius norm of the Jacobian matrix, which we will refer to as Jacobian Norm as a short hand. We report two measures based on the Jacobian:

1. Jacobian: The first measure computes the Jacobian norm on the held out set $\mathcal{T}_{\text{te}}$. From (Novak et al., 2018) higher norm intuitively means the point is dissimilar to the training distribution, and thus we should expect worse generalization.

2. Jacobian-Diff: This measure is similar to the Jacobian measure, except we use the training data $\mathcal{S}$ to provide a baseline for the test Jacobian. It is computed as the difference between the Jacobian for source and target. We also consider measures based on the ratio of the Jacobians and the log of the ratios of the Jacobians. We call these measures Jacobian-Ratio and Jacobian-Log-Diff, respectively.

**Mixup Based Measures.** Mixup (Zhang et al., 2018) was proposed as a more robust alternative to Empirical Risk Minimization, where given two examples $X_1$, $X_2$, and associated labels $y_1$, $y_2$, one optimizes an objective that samples $\lambda \sim Beta(\alpha, \alpha)$ and feeds the learning machine inputs $\tilde{X} = \lambda \cdot X_1 + (1 - \lambda) \cdot X_2$ and targets $\tilde{y} = \lambda \cdot y_1 + (1 - \lambda) \cdot y_2$. Given $\{\tilde{X}, \tilde{y}\}$, one proceeds as if one were doing standard Empirical Risk Minimization. It is easy to see that Mixup encourages the learned function to be smooth, and indeed has been shown to smooth the input-output Jacobian (Carratino et al., 2020) of the network. Here, we adapt the mixup idea not as a training algorithm but as a generalization measure, and use the model's score function $c$ instead of the labels $y$ for interpolation. Our intuition is the same as that for the Jacobian, namely, that if the function is not smooth around target examples, the network should not generalize as well. Given the neural network function $c(X)$, and a dataset $\mathcal{T}$, and $\lambda \sim Beta(\alpha, \alpha)$, we compute the Mixup measure as:

$$\frac{1}{|\mathcal{T}|} \sum_{X_i \in \mathcal{T}, X_j \in \mathcal{T}; i \neq j} (\lambda c(X_i) + (1 - \lambda)c(X_j)$$
$$- h(\lambda X_i + (1 - \lambda)X_j)^2 \tag{10}$$

We experiment with two values of $\alpha \in \{0.1, 0.3\}$ following (Zhang et al., 2016). Similar to the Jacobian, we use a relative variant of Mixup, namely Mixup-Diff which computes the difference of the Mixup between $\mathcal{T}_{\text{tr}}$ and $\mathcal{S}_{\text{tr}}$, and another variant which computes the log of the difference (Mixup-log-Diff) between $\mathcal{T}_{\text{tr}}$ and $\mathcal{S}_{\text{tr}}$.

We then compute the following measures with mixup, setting $\alpha = 0.1$:

- Mixup: compute $\mu_{\mathcal{T}_{\text{tr}}}$

- Mixup-Diff: compute $\mu_{\mathcal{T}_{\text{tr}}} - \mu_{\mathcal{S}_{\text{tr}}}$

- Mixup-Ratio: compute $\frac{\mu_{\mathcal{T}_{\text{tr}}}}{\mu_{\mathcal{S}_{\text{tr}}}}$

- Mixup-log-Diff: compute $\log\left(\mu_{\mathcal{T}_{\text{tr}}}\right) - \log\left(\mu_{\mathcal{S}_{\text{tr}}}\right)$

We also compute each of the above measures with $\alpha = 0.3$, yielding measures Mixup-Alpha-0.3 and so on.

---

**Algorithm 6** Mixup for a dataset $\mathcal{U}$

---

1: **procedure** PERMUTEMINIBATCH($\mathcal{B}$)
2:    $\mathcal{I} \leftarrow$ permute $((1, \cdots, |\mathcal{B}|))$
3:    **return** $(\mathcal{B}_i$ for $i \in \mathcal{I})$
4: **end procedure**

1: **procedure** MIXUP($\mathcal{U}, \hat{c}, \alpha$)
2:    $\mu \leftarrow 0$
3:    $t \leftarrow 0$ Minibatch $\mathcal{B} \in \mathcal{U}$
4:    $\tilde{\mathcal{B}} \leftarrow$ PermuteMinibatch($\mathcal{B}$) $X_1 \in \mathcal{B}$ and $X_2 \in \tilde{\mathcal{B}}$
5:    $\lambda \sim \text{Gamma}(\alpha, \alpha)$
6:    $A \leftarrow \lambda \hat{c}(X_1) + (1 - \lambda)\hat{c}(X_2)$
7:    $B \leftarrow \hat{c}(\lambda X_1 + (1 - \lambda)X_2)$
8:    $\mu \leftarrow \mu + ||A - B||_2^2$
9:    $t \leftarrow t + 1$
10:    **return** $\frac{\mu}{t}$
11: **end procedure**

---

**Entropy on Source Data.** We compute the output-entropy of the neural network $h$ on the source domain data $\mathcal{S}_{tr}$ (Entropy-Source):

$$\frac{1}{|\mathcal{S}_{\text{tr}}|} \sum_{X_i \in \mathcal{S}_{\text{tr}}} \sum_{j=1}^{\mathcal{Y}} -\log\left(c(X_i)[j]\right) \cdot c(X_i)[j]. \tag{11}$$

**Entropy on Target Data.** We also compute the output-entropy of the neural network $h$ on the target domain data $\mathcal{T}_{tr}$ (Entropy-Target):

$$\frac{1}{|\mathcal{T}_{\text{tr}}|} \sum_{X_i \in \mathcal{T}_{\text{tr}}} \sum_{j=1}^{\mathcal{Y}} -\log\left(c(X_i)[j]\right) \cdot c(X_i)[j]. \tag{12}$$

# 8. Implementation Details

We first explain the exact model architecures we use for each dataset in the paper, then provide more details on the hyperparameter choices, and finally provide more details on relevant hyperparameters for computing the generalization measures.

## 8.1. Network architectures

We use the same architecture as that used in DomainBed (Gulrajani & Lopez-Paz, 2020) for the RotatedMNIST dataset (fig. 4). As mentioned in the main paper, this model has 386K parameters. For PACS and VLCS we use standard ResNet50 models pretrained on ImageNet. For this we use the standard, available implementations in the PyTorch (Paszke et al., 2019) library.

## 8.2. Random Hyperparameter Sweep

As explained in the main paper, we perform a random hyperparameter sweep to obtain a set of ERMs which generalize to different degrees. Here we describe in more detail how we pick the hyperparameters.

We first describe the hyperparameter search attributes for RotatedMNIST. `Uniform(X, Y)` denotes a uniform distribution in the continuous interval `(X, Y)`, while `Uniform[X, Y]` denotes a discrete choice between elements X and Y. `pow(X, Y)` denotes $X^Y$. Given this notation, our hyperparameter choices for RotatedMNIST are:

- Learning Rate: `pow(10, Uniform(-4.5, -2.5))`

- Batch Size: `pow(2, Uniform(3, 9))`

Next, our hyperparameter choices for VLCS and PACS are:

- Learning Rate: `pow(10, Uniform(-5, -3.5))`

- Batch Size: `pow(2, Uniform(3, 5.5))`

- Dropout: `Uniform[0, 0.1, 0.5]`

- Weight Decay: `pow(10, Uniform(-6, -2))`

## 8.3. Hyperparameters for Fisher

Our computation of the Fisher approximates the true Fisher information by computing it over a subset of data examples $\tilde{N}$ (as explained in the main paper). We use $\tilde{N} = 75$ for PACS and VLCS and $\tilde{N} = 1000$ for RotatedMNIST.

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

```
        (conv1): Conv2d(3, 64, kernel_size=(3, 3), stride=(1, 1), padding=(1, 1))
         (Relu): ReLU()
          (bn0): GroupNorm(8, 64, eps=1e-05, affine=True)

        (conv2): Conv2d(64, 128, kernel_size=(3, 3), stride=(2, 2), padding=(1, 1))
         (Relu): ReLU()
          (bn1): GroupNorm(8, 128, eps=1e-05, affine=True)

        (conv3): Conv2d(128, 128, kernel_size=(3, 3), stride=(1, 1), padding=(1, 1))
         (Relu): ReLU()
          (bn2): GroupNorm(8, 128, eps=1e-05, affine=True)

        (conv4): Conv2d(128, 128, kernel_size=(3, 3), stride=(1, 1), padding=(1, 1))
         (Relu): ReLU()
          (bn3): GroupNorm(8, 128, eps=1e-05, affine=True)

      (avgpool): AdaptiveAvgPool2d(output_size=(1, 1))
 (squeezeLastTwo): SqueezeLastTwo()
```

Figure 4: **Schematic illustration of forward pass of MNIST CNN**. We list the modules which are executed on the input image (using pytorch classes (Paszke et al., 2019)). Modules listed at the top are executed first, followed by each module in the sequence. `SqueezeLastTwo` drops the last two dimensions of the tensor from the previous layer.

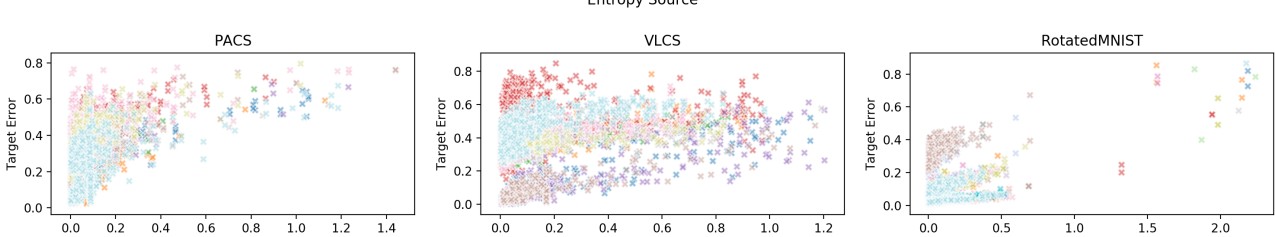

Figure 5: $\epsilon_T(\hat{c})$ *v.s.* **Measure for different datasets**. Each environment for a given dataset has a marker of a different color. The measure being computed is listed at the in the title of the figure

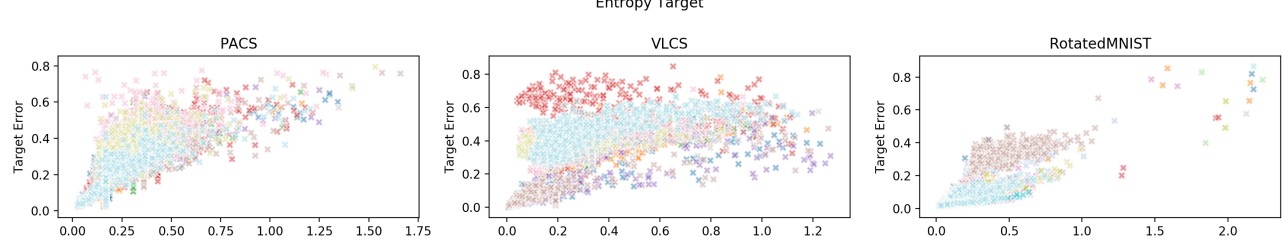

Figure 6: $\epsilon_T(\hat{c})$ *v.s.* **Measure for different datasets**. Each environment for a given dataset has a marker of a different color. The measure being computed is listed at the in the title of the figure

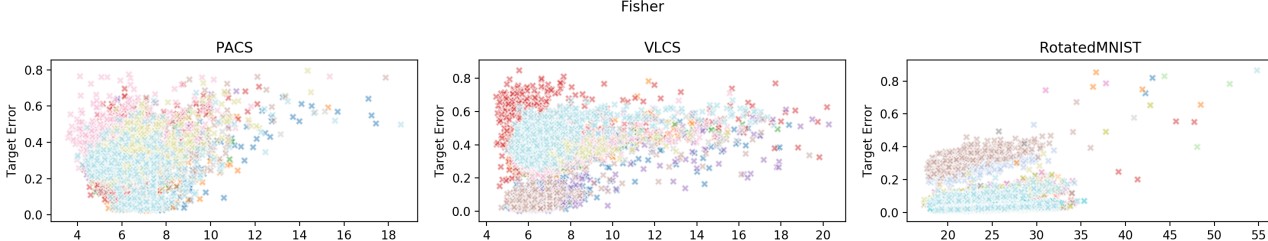

Figure 7: $\epsilon_T(\hat{c})$ *v.s.* **Measure for different datasets**. Each environment for a given dataset has a marker of a different color. The measure being computed is listed at the in the title of the figure

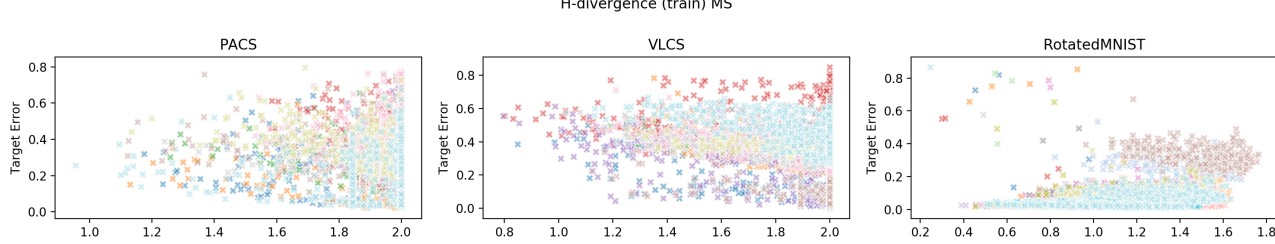

Figure 8: $\epsilon_T(\hat{c})$ *v.s.* **Measure for different datasets**. Each environment for a given dataset has a marker of a different color. The measure being computed is listed at the in the title of the figure

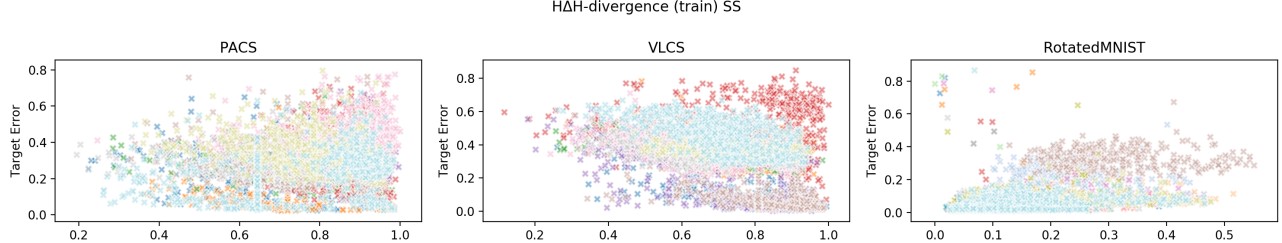

Figure 9: $\epsilon_T(\hat{c})$ *v.s.* **Measure for different datasets**. Each environment for a given dataset has a marker of a different color. The measure being computed is listed at the in the title of the figure

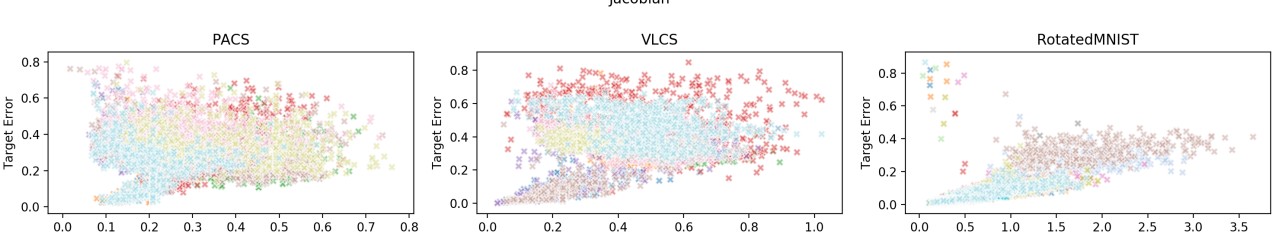

Figure 10: $\epsilon_T(\hat{c})$ *v.s.* **Measure for different datasets**. Each environment for a given dataset has a marker of a different color. The measure being computed is listed at the in the title of the figure

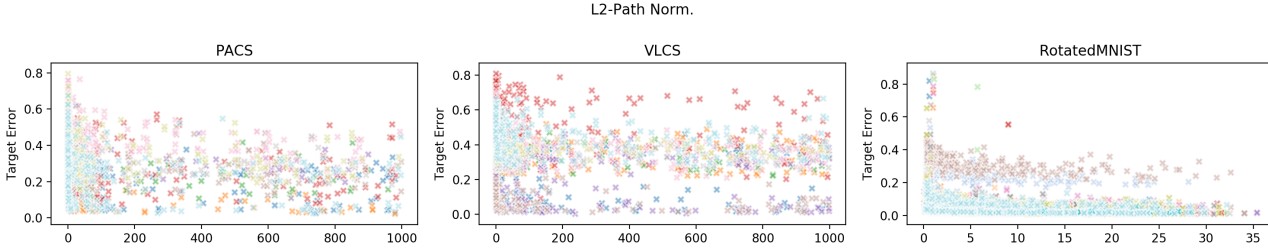

Figure 11: $\epsilon_T(\hat{c})$ *v.s.* **Measure for different datasets**. Each environment for a given dataset has a marker of a different color. The measure being computed is listed at the in the title of the figure

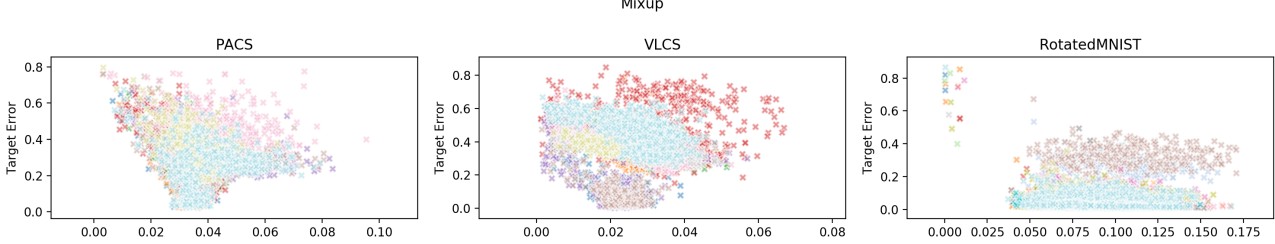

Figure 12: $\epsilon_T(\hat{c})$ *v.s.* **Measure for different datasets**. Each environment for a given dataset has a marker of a different color. The measure being computed is listed at the in the title of the figure

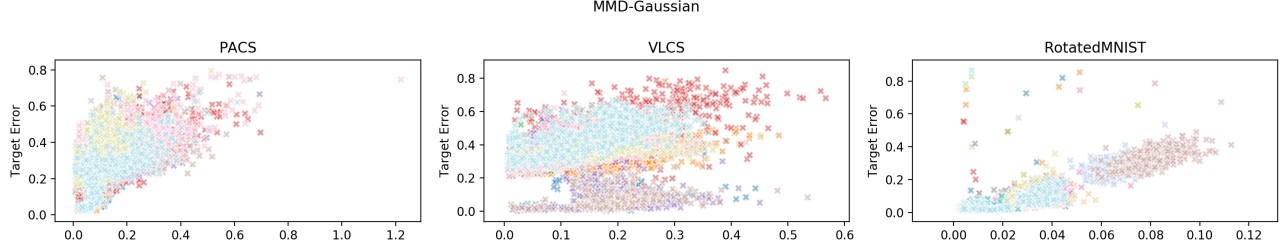

Figure 13: $\epsilon_T(\hat{c})$ *v.s.* **Measure for different datasets**. Each environment for a given dataset has a marker of a different color. The measure being computed is listed at the in the title of the figure

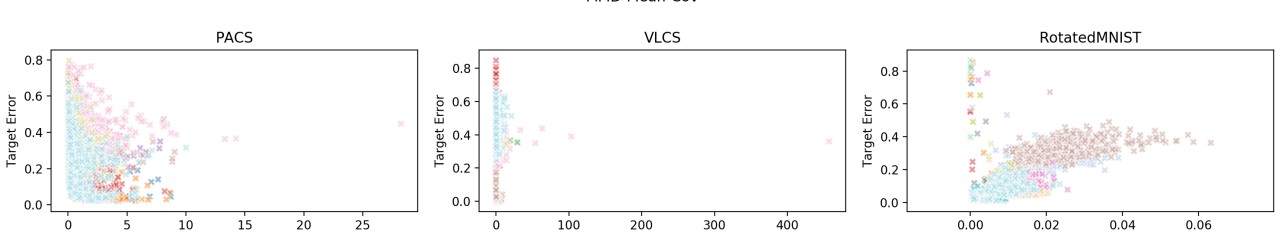

Figure 14: $\epsilon_T(\hat{c})$ *v.s.* **Measure for different datasets**. Each environment for a given dataset has a marker of a different color. The measure being computed is listed at the in the title of the figure

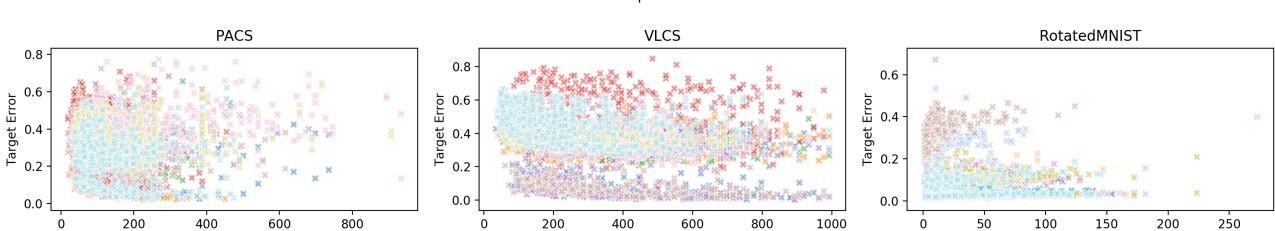

Figure 15: $\epsilon_T(\hat{c})$ *v.s.* **Measure for different datasets**. Each environment for a given dataset has a marker of a different color. The measure being computed is listed at the in the title of the figure

from different domains. *Machine Learning*, 79:151–175, 2010. URL http://www.springerlink.com/content/q6qk230685577n52/.

Carratino, L., Cissé, M., Jenatton, R., and Vert, J.-P. On mixup regularization. June 2020.

Gulrajani, I. and Lopez-Paz, D. In search of lost domain generalization. July 2020.

Hoffman, J., Roberts, D. A., and Yaida, S. Robust learning with jacobian regularization. August 2019.

Jiang, Y., Neyshabur, B., Mobahi, H., Krishnan, D., and Bengio, S. Fantastic generalization measures and where to find them. December 2019.

Kuhn, H. W. The hungarian method for the assignment problem. *Nav. Res. Logist. Q.*, 2(1-2):83–97, March 1955.

Lopez-Paz, D. and Oquab, M. Revisiting classifier two-sample tests. *arXiv preprint arXiv:1610.06545*, 2016.

Novak, R., Bahri, Y., Abolafia, D. A., Pennington, J., and Sohl-Dickstein, J. Sensitivity and generalization in neural networks: an empirical study. February 2018.

Paszke, A., Gross, S., Massa, F., Lerer, A., Bradbury, J., Chanan, G., Killeen, T., Lin, Z., Gimelshein, N., Antiga, L., Desmaison, A., Köpf, A., Yang, E., DeVito, Z., Raison, M., Tejani, A., Chilamkurthy, S., Steiner, B., Fang, L., Bai, J., and Chintala, S. PyTorch: An imperative style, High-Performance deep learning library. December 2019.

Sun, B. and Saenko, K. Deep CORAL: Correlation alignment for deep domain adaptation. July 2016.

Zhang, C., Bengio, S., Hardt, M., Recht, B., and Vinyals, O. Understanding deep learning requires rethinking generalization. November 2016.

Zhang, H., Cisse, M., Dauphin, Y. N., and Lopez-Paz, D. mixup: Beyond empirical risk minimization. February 2018.