# OpenReview forum: "An Empirical Investigation of Domain Generalization with Empirical Risk Minimizers"
_NeurIPS.cc/2021/Conference — NeurIPS 2021 Poster_

### Official Review · Reviewer_Dvcv · 2021-07-16

**Rating:** 6
**Confidence:** 4

**Summary:**

This paper conducts a large-scale empirical study of ERM models tested in the domainBed pipeline, together with the empirical results tested, the authors also investigated the numerical values of several different measures, which allows the authors to further study the relationship between multiple different measures and the empirical performances. The empirical study suggests that there may be interesting connections between the target-entropy and the empirical performances. The paper candidly acknowledges its limitations of being purely empirical.

**Limitations And Societal Impact:**

Yes.

**Main Review:**

The paper offers a large-scale empirical study aiming to discuss the relationship between several measures of the models and the empirical performance, across a large range of models as well as benchmarks. The paper offers interesting insights of several measures. Overall, I deem it a valuable input for the community, but I have several concerns/questions:

1. I appreciate that the authors have listed their improvements upon previous submissions and particularly mention that the paper now has a different narrative/story/motivation. However, unfortunately, I think the narrative can be further improved. In particular, it is not a surprise that Ben-David theory has limitations in predicting domain generalization performance and multiple previous papers have discussed it, such as [1,2,3], it seems not fruitful that the authors put another evidence to show that Ben-David theory has flaws in practice.
    * Alternatively, I wonder if the narrative of evaluating the relationship between different measures, while treating all the measures in the same manner has been considered previously by the authors or reviewers. I consider this narrative much more scientific. There seems no particular need to emphasize the limitations of Ben-David's theory given previous literature.

2. It is interesting that Entropy-Target stands out as a criterion in Table 1. What's the relationship between Entropy-target, entropy-source, and source error? Is there a chance that source error is related to entropy-source, (and further entropy-target), thus the good relationship between entropy-target is actually a confounded relationship between source error and target error?
    * It might be better to also list the pair-wise correlation between all these criteria listed and see if all these criteria are independent. Otherwise, it will be misleading that any criteria stand out while also highly correlated with source error.
    * I wonder if there is any way to calculate the correlations conditioning on the source error.

3. I do not get the paragraph at line 266 on "What contributes to the performance of Entropy-Target". I don't see how this paragraph answers the question other than showing another specific example that entropy and performance are correlated in the two test scenarios 15 vs. 0. It seems the authors aim to offer an investigation of the reason, but the discussion still, unfortunately, didn't go any further than describing the empirical performance. In particular, any conjectures on why higher confidence and better generalization cooccurs? (BTW, it might be better to define confidence here).

4. Minorly, at line 236, "intuitively, a model that is less sensitive to changes in the input will generalize better", I wonder if there any evidence supporting this discussion.

[1] On Learning Invariant Representations for Domain Adaptation

[2] Domain Adaptation with Asymmetrically-Relaxed Distribution Alignment

[3] Learning Robust Models by Countering Spurious Correlations (unpublished work, proceed with cautious)

**Time Spent Reviewing:**

2.5

---

> ### Author Response · Authors · 2021-08-10
> **Thanks for the valuable feedback! Initial response below.**
>
> We thank the reviewer for the detailed review and are glad they thought the work offered interesting insights of different measures, and is a valuable input for the community. We address below the specific concerns and hope to engage in any further discussion to help address the concerns of the reviewer.
>
> * **“Why do we need a new datapoint given [1, 2, 3] showing theory is not useful in practice?”**
>
> While the references [1, 2] pointed out by the reviewer consider the setup of domain adaptation (unlabeled data from the target domain used during training), we study the different scenario of domain generalization (no information about the target domain used during training).
>
> Interestingly, training ERM models under the domain generalization setup reveals a new case where Ben-David’s bounds may be of limited practical utility [a]. More specifically, these bounds sum three terms, namely (i) the training error, (ii) the distributional discrepancy between the source and target domains, and (iii) the lambda closeness (i.e. compatibility of decision boundary). While in domain generalization we do not have any control over the distributional discrepancy, we observe that this term does not accurately predict target performance if the decision boundary stays the same. To exemplify, picture two Gaussians separated by the same linear decision boundary (where the * and & domains each have classes 1 and 2):
>
>              *1  | *2
>              $1  | $2
>
> Since the decision boundary stays the same (meaning the decision boundaries are compatible) but the distributional discrepancy is large (with respect to H-divergence for a linear family; the domains could be separated by a horizontal line), we simultaneously obtain a large Ben-David bound but a small target error. This sort of slack in the bound is the kind of failure mode for its predictive utility elucidated by our domain generalization experiments.
>
> In contrast, previous works [1, 2] consider the different, domain adaptation (not generalization) problem where minimizing the distributional discrepancy (term (ii) of the Ben-David bound) -- which requires utilization of the target input data -- might result in a large value for the third term (making the decision boundaries incompatible). Finally, as far as we could understand, reference [3] talks about using additional domain knowledge about spurious features to obtain tighter bounds than Ben-David et. al., which is an interesting but orthogonal problem to our setting where we would like to utilize the theory as a diagnostic to reveal which ERMs generalize in domain generalization (where such side information is not readily available).
>
> Thus, we think our work offers a different perspective on the practical utility of the bounds in a different problem setting compared to previous work. However, we will be happy to include a discussion of this form in the paper and thank the reviewer for bringing these works to our attention.
>
> * **“Relationship between Entropy-Target, Entropy-Source and Source Error”**:
>
> L278-282 discusses the phenomenon that the reviewer is concerned about, namely that Entropy-Source is highly related to the source error and that is why it performs so well. On the other hand, Entropy-Target does not seem to exhibit the same pathology, since when added to the Source Error as a feature in the Joint setting (Table 1, row 4, column “Joint”) it achieves a Spearman’s Rho of 0.77 which is higher than Source Error alone at 0.71. Note that in contrast, Entropy-Source does not improve upon the Source Error in the same setting (Table 1, row 3, column “Joint”), suggesting strong dependency between it and the Source Error.
>
> **“Conditioning on source error”**:
> We believe studying any improvements over the source error’s performance when used in conjunction with the source error is a good measure of “conditioning” on the source error and looking at how much of the residual variation is explained by a given measure. This is in fact the methodology we used to bold the entries in Table 1 (as explained in the caption). We will highlight this more clearly in the revised version.
>
> **“Explore pairwise correlations”**:
> Appendix Fig. 3 indeed plots the pairwise correlations between all the measures and can be used to dig deeper into the dependencies between the various measures as the reviewer suggested. We will add a pointer to this figure in the main text.
>
> * **Why does Entropy-Target do well? -- Text in the paper is not clear.**
>
> We were trying to relate the discussion in this section to Fig. 1 in the paper. Essentially, Entropy-Target is able to distinguish well if an image has 0 degree rotation or 15 degree rotation (as shown in Fig. 4). Thus, it achieves a high correlation since Fig. 1 shows that the performance difference between 0 and 15 degree rotations is substantial (forming two different clusters) when trained on 30-75 degree rotations. Essentially, in this regime one can achieve good correlation with generalization by recognizing whether a digit is upright or 15 degree rotated which Entropy-Target seems to do well. We will ground the discussion in that section with Fig. 1 to make it more clear.
>
> * **Minorly, at line 236, "intuitively, a model that is less sensitive to changes in the input will generalize better", I wonder if there any evidence supporting this discussion.**
>
> Novak et.al. (cited in L236) is a good reference for experimental evidence supporting the statement.
>
> *References:*
> [a] A theory of learning from different domains
> S Ben-David, J Blitzer, K Crammer, A Kulesza, F Pereira, JW Vaughan
> Machine learning 79 (1), 151-175 (2010)

---

> > ### Comment · Reviewer_Dvcv · 2021-08-11
> > **Continued Discussion for Initial Response**
> >
> > I would like to thank the authors for their prompt and detailed responses. The response mostly addressed my concerns, but I still have a couple more questions.
> >
> > **About the additional discussions**
> >
> > > *While the references [1, 2] pointed out by the reviewer consider the setup of domain adaptation (unlabeled data from the target domain used during training), we study the different scenarios of domain generalization (no information about the target domain used during training).*
> >
> > I agree, but I think the differences between DA and DG are not so much that the working challenging DA can be safely ignored. In fact, the author also explicitly said in the paper (line 52-53)
> > >*Therefore, to remain in the domain generalization setup, we will put the theory of Ben-David et al. (2007) to use from a different angle.*
> >
> > which suggests the authors agree that works on DA can help offer to understand works for DG.
> > In addition, to quote the author's response:
> > >*this term does not accurately predict target performance*
> >
> > is what I consider the high-level message from [1,2,3], all from different perspectives, as the authors listed above. I agree the author is discussing this problem from a new, potentially orthogonal perspective, my point is that including them can further complete this paper's discussion even though they sit on the empirically distinct, but theoretically not-so-distinct (as agreed by the authors) setup.
> >
> > **What contributes to the performance of Entropy-Target**
> >
> > Thanks for the explanation. However, my point is this title "What contributes to the performance of Entropy-Target" suggests a much deeper investigation, even with a theoretical conclusion, other than this, current, discussion centering around the empirical performances. The authors' response suggests that this current form of discussion is what they intend to do. In this case, I will suggest the authors update this paragraph title. Same for the title at line 280.
> >
> > **About the reference required**
> > >*Novak et.al. (cited in L236) is a good reference for experimental evidence supporting the statement.*
> >
> > This is not true in the current PDF. Novak et.al. is mentioned at L238 after another sentence, it's not so clear that it was used to support this claim. Did the response mean the authors intend to move it to L236 or something else?

---

> > > ### Author Response · Authors · 2021-08-13
> > > **Response to Continued Discussion**
> > >
> > > We thank the reviewer for their prompt responses and for engaging in discussion.
> > >
> > > **“Inapplicability of Theory”**: We certainly agree with the reviewer (and our statement in the paper) that the theory of Domain Adaptation (DA) is applicable to Domain Generalization (DG). In our initial response, we emphasized that the failure mode when applying the theory is different based on the setting of DA or DG. In the case of adaptation, the “failure” is because the third term in the theory (lambda closeness) becomes too large, while in our observation the “failure” in generalization is because of the second term (distributional discrepancy). Thus, the different task settings result in different kinds of failure modes for the practical application of theory.
> > >
> > > We agree that adding a discussion explaining these nuances, and crediting previous works for their role in identifying these other distinct issues with the theory, will indeed improve the paper. We will be happy to add a paragraph reflecting this to the main paper, and thank the reviewer for engaging in discussion which brought some of these nuances to light.
> > >
> > > **What contributes to the performance of Entropy-Target**
> > > We can see how the current title could be misunderstood as overclaiming an understanding we don’t have, and we will modify the title to “Practically analyzing the performance of Entropy-Target”.
> > >
> > > **About the reference required**
> > > We will add the reference to L236 and modify it to “Intuitively, a model that is less sensitive to changes in the input will generalize better, as also observed experimentally in (Novak et.al., 2018).” Thanks for pointing this out!

---

> > > > ### Comment · Reviewer_Dvcv · 2021-08-13
> > > > **Thanks for the response**
> > > >
> > > > I have updated my ratings.

---

### Official Review · Reviewer_7pTi · 2021-07-17

**Rating:** 7
**Confidence:** 3

**Summary:**

Prior work has found that empirical risk minimization performs unexpectedly well on domain generalization tasks. In this work, the authors try to explain this phenomenon. First, they use theory developed for domain adaptation to bound the target domain test error, finding that theory-based measures do not adequately predict the target domain test error, especially when target domain labels are not available. Next, the authors try a variety of empirical measures. They find that some measures outperform the theory, especially when combined with the in-domain test error.

**Limitations And Societal Impact:**

Yes, the authors adequately addressed the limitations and potential negative societal impact of their work.

**Main Review:**

Strong points

- The paper is clearly written and relevant to those working in the domain generalization field.
- The authors are rigorous in testing both theory-based measures and a comprehensive list of empirical measures.


Potential Improvements
- I would recommend splitting Table 1 to present separate numbers for each dataset. This would also allow some discussion of which measures performs well on which dataset and why.
- I would recommend the authors also run their analyses on ColoredMNIST, as 1) it is the standard dataset used to benchmark most domain generalization methods, 2) it is one dataset where IRM actually outperforms ERM for some model selection strategies, and 3) it is a dataset where spurious correlation is simple and synthetically added, as opposed to the three datasets currently examined in the paper.
- Figure 3 is a bit confusing to me - the authors should define how looseness is calculated (I'm assuming it's the bound minus the target domain error), and the equation reference on line 175 is also missing. In addition, if $\epsilon_S(\hat{h})$ is the same across divergence measures, I would recommend making it a single bar for clarity.
- I would recommend moving the full equations for the bounds from Appendix Section 7 into the main paper Section 2.
- The authors could consider strengthening their analyses by adding a couple of additional datasets (maybe from non-image modalities) from the WILDS paper [1], which has also shown that ERM outperforms IRM on domain generalization tasks. The datasets proposed in WILDS also tend to be more practically useful than the ones from DomainBed. I do realize that this would be quite time-consuming though.

Recommendation

I recommend acceptance of this paper, since I believe it is a valuable continuation of the discussion started by the DomainBed paper, and the insights from this paper could lead to additional theoretical or methodological developments in domain generalization.

[1] https://arxiv.org/pdf/2012.07421.pdf

**Time Spent Reviewing:**

4

---

> ### Author Response · Authors · 2021-08-10
> **Thanks for the valuable feedback! Addressing specific points of improvement below.**
>
> We thank the reviewer for their detailed feedback and are glad they find this paper to be a valuable contribution. We respond to the specific comments below:
> 1. **Per-dataset measures**: Fig. 2 (Appendix) shows the results of each of the measures per dataset. We will move that figure to the main paper for the revised version of the paper and add a discussion of the differences between datasets.
> 2. **Adding ColoredMNIST**: We initially did not include ColoredMNIST since the labeling function changes (adversarially) in some of the environments, meaning that in the pixel domain, the domains would not be lambda close (with respect to the theory of Ben-David et. al.). We will, however, add results on ColoredMNIST for completeness in the revision of the paper.
> 3. **How is looseness defined?**: Looseness is defined as the difference between the left hand side of Eqn. 1 (Appendix 7) and the right hand side of Eqn. 1. Namely, it is the difference between the observed target domain error, and the sum of the source domain error, divergence between the domains, and lambda closeness. We will clarify this in the revised version.
> 4. **Adding WILDS dataset**: Adding a dataset from the WILDS benchmark. We will add results on WILDS Camelyon17 (Tissue images) to the revised version of the paper. Adding non-image based datasets is interesting, but complicates / makes some measures difficult to compute, for example the input output Jacobian, due to discreteness of input. While we think this would be quite interesting to study, it is out of scope for this paper.
> 5. **Eqn. 1 to main paper**: We will move Eqn. 1 (Appendix) to the main paper. Thanks for the suggestion!
> 6. **Improvements to Fig.3**: Making source domain error a single bar -- this is a great point! We will make the change for the updated version of the paper.

---

### Official Review · Reviewer_mk3V · 2021-07-18

**Rating:** 6
**Confidence:** 3

**Summary:**

This work tried to find that which measures can be used for predicting the the out-of-domain generalization for the deep neural networks trained using Empirical Risk Minimization (ERM). The authors started from the domain adaptation theory of Ben-David, 2007 but found that it had limited ability. Then they explored many other measures, including Fisher based measures, Jacobian Norm based measures, Mixup based Measures, and entropy on target data. They found the best single factor led to around 0.70 Spearman's ρ and the joint use can boost the value to 0.80.

**Limitations And Societal Impact:**

Yes.

**Main Review:**

1. I think it is a valuable work, given its scale for both datasets and different measures.

2. Perhaps a bit frustratingly, none of the tested measure showed a strong correlation for OOD performance.

3. It could help the community further if the authors can release the statistics of their 12,000 models, as not many researchers have the resources to repeat those.

**Time Spent Reviewing:**

2

---

> ### Author Response · Authors · 2021-08-10
> **Thanks for the feedback! Addressing specific concerns below.**
>
> We are glad the reviewer thought that the work was valuable. We address specific concerns below, and hope to engage in further discussion as required:
>
> 1. **None of the measures show strong correlation**: In the Stratified setting (Table 1) we emphasize that it is possible for measures to improve to as high as 0.83 Spearman’s rho. Moreover, measures like the Jacobian and Entropy-Target seem promising in the Joint setting. Thus, we see promise for measuring generalization from the measures considered in the paper, but at the same time agree that significantly more work needs to be done on this problem.
>
> 2. **Statistics of 12,000 models**: We will release at least a table with all the statistics of the models and the different measures computed on them, and are actively exploring how to share the model weights (which are of the order of 1TB on disk) upon publication, so that the community can explore other measures.

---

### Official Review · Reviewer_N5g2 · 2021-08-01

**Rating:** 7
**Confidence:** 3

**Summary:**

The paper provides a large scale study of out-of-distribution generalization measures. It first provides a brief introduction of the core theoretical foundations of the field. Then the paper trains 12000 models on DomainBed Benchmarks to examine the predictive power of theory-based measures. The paper argues that theory based measure fail to accurately capture the OOD generalization behavior. Following the methodology of Jiang 2019 et al, the paper examine how various empirical measures (sharpness, entropy, Fisher, ...) predict OOD generalization behavior. In the end, the paper lists the predictive measures and examines their performance details.

**Limitations And Societal Impact:**

adequately addressed

**Main Review:**

The paper is written well and it tackles an interesting and fundamental problem. I list my comments and questions below:

- Figure 2 is rather valuable as it contains the summary of your empirical results. Unfortunately, in its current format, it is rather hard to read. Moreover, as legends are not provided with the figure, interpreting the figure is also hard.

- The bounds provided by Ben-David et al are one of the main subjects of the paper. However, no stand alone equations showing the bounds are provided. Line 175 attempts to reference one such equation but it seems that the equation was removed.

- Figure 3 is very interesting. I had a hard time interpreting the results as no mathematical expression provided on how looseness is calculated.

- Subsection 3.3 feels a little out of place. As it describes the methodology used in Section 5, maybe it can be placed there?
-Minor typos in line 222

**Time Spent Reviewing:**

3

---

> ### Author Response · Authors · 2021-08-10
> **Thanks for the feedback! Addressing concerns below.**
>
> We thank the reviewer for valuable feedback! We are glad the reviewer thought our paper tackles an interesting and fundamental problem. We respond to each of the comments below:
>
> 1. **Readability of Fig. 2**: As explained in the caption, each of the points in Fig. 2 corresponds to a trained model, and all models trained and evaluated on the same pairs of (train, test) environments (which we call conditions) are labelled with the same color. There are numerous conditions for each dataset, which makes adding a legend difficult. We will illustrate more clearly what each dot means by linking Fig. 1 with Fig. 2 and overlaying an example on the figure to illustrate it better. Thanks for the suggestion to improve this Figure.
> 2. **Equations for bounds not given**: Appendix 7 shows the relevant equations for the bounds. We will add an equation similar to Appendix 7 (Eqn. 1) to the main paper.
> 3. **How is looseness defined?** Looseness is defined as the difference between the left hand side of Eqn. 1 (Appendix 7) and the right hand side of Eqn. 1. Namely, it is the difference between the observed target domain error, and the sum of the source domain error, divergence between the domains, and lambda closeness. We will clarify this in the text.
> 4. **Flow improvements**: We will move Subsection 3.3 to Sec. 5 — thanks for the suggestion, we agree it will improve the flow!

---

### Decision · Program_Chairs · 2021-09-27

**Decision:**

Accept (Poster)

**Comment:**

The authors perform a large-scale empirical evaluation of how well empirical risk minimisers perform out-of-domain generalisation, and how well this performance is predicted by various properties of the source and target domains. They find that measures of domain discrepancy which have been used to bound domain adaptation error were not among the most predictive aspects, but that other measures showed good correlation with target error.

The reviewers all recognised the value of an empirical study like this, and were largely happy with how it was conducted. Reviewer Dvcv questioned the strong focus on the theory of Ben-David, and how it was a surprisingly poor predictor, when several more recent works have already pointed out the looseness in those early results. Additionally, several reviewers asked for increased clarity surrounding some of the results. Nevertheless, I believe this work is likely to lead to new theoretical insights and empirical results.